# Strengthening in multi-principal element alloys with local-chemical-order roughened dislocation pathways

Qing-Jie Li[1], Howard Sheng[2,3] & Evan Ma [1]

High-entropy and medium-entropy alloys are presumed to have a configurational entropy as high as that of an ideally mixed solid solution (SS) of multiple elements in near-equal proportions. However, enthalpic interactions inevitably render such chemically disordered SSs rare and metastable, except at very high temperatures. Here we highlight the wide variety of local chemical ordering (LCO) that sets these concentrated SSs apart from traditional solvent-solute ones. Using atomistic simulations, we reveal that the LCO of the multi-principal-element NiCoCr SS changes with alloy processing conditions, producing a wide range of generalized planar fault energies. We show that the LCO heightens the ruggedness of the energy landscape and raises activation barriers governing dislocation activities. This influences the selection of dislocation pathways in slip, faulting, and twinning, and increases the lattice friction to dislocation motion via a nanoscale segment detrapping mechanism. In contrast, severe plastic deformation reduces the LCO towards random SS.

---

[1] Department of Materials Science and Engineering, Johns Hopkins University, Baltimore, MD 21218, USA. [2] Department of Physics and Astronomy, George Mason University, Fairfax, VA 22030, USA. [3] Center for High Pressure Science and Technology Advanced Research, 201203 Shanghai, China. Correspondence and requests for materials should be addressed to H.S. (email: hsheng@gmu.edu) or to E.M. (email: ema@jhu.edu)

Multi-principal element materials, with compositions in the central region of the multicomponent phase diagram, are currently dubbed as high-entropy alloys (HEAs). These complex concentrated alloys are emerging as a new research field attracting considerable attention[1–11]. However, a fundamental materials science question remains outstanding in the community, namely, what is the new tell-tale feature that distinguishes these HEAs from traditional solid solution (SS) alloys. The original answer was that the HEAs correspond to an unusually high configurational entropy of mixing ($S_c$)[1], with a magnitude >~1.61R estimated from $S_{c,ideal} = -R \sum_{i=1}^{N} x_i \ln(x_i)$ for ideal solutions, where $R$ is the gas constant and $x_i$ is the molar fraction of the $i$th component. As of today, such a random SS (RSS) picture remains a common assumption in the HEA community. However, this RSS state is possible only at very high temperatures, where the degree of local chemical order (LCO) is negligible and the entropy term predominates the free energy reduction to dictate ideal mixing. Real-world HEAs are actually processed (annealed and homogenized) and used at relatively low temperatures[10,12–18], where complex enthalpic interactions among various constituent elements come into play. Even the pioneering Cantor alloy, a face-centered cubic (FCC) SS close to random if annealed at 1100 °C[19], decomposes after long anneal at <900 °C[17]. Appreciable chemical order has in fact been found in a number of HEAs[20–27]. As such, HEAs, even when in the form of a single-phase SS, can be of a mixing entropy far below $S_{c,ideal}$.

Considering the metastable nature of HEAs, we advocate a different perspective: they are special in the vastness of possible LCO configurations, and as such even a given HEA composition can deliver a plethora of properties. Specifically, a concentrated SS ushers in an unprecedentedly large variability of LCOs, from the extreme of (overly simplified) RSS, all the way to fully ordered ground-state intermetallics. These intermediate states highlight HEA's metastable nature, and are beyond reach in traditional (terminal) SSs, which can instead be approximated as Raoult's or Henry's solution, where solutes always sparsely and randomly distribute in a solvent crystal lattice, producing only one specific set of properties. Now for concentrated HEAs, the type (species involved), degree (magnitude of the order parameter), and extent (length scale and spatial distribution) of LCO all span a wide range.

In what follows we systematically demonstrate, using an atomistic model that mimics the FCC NiCoCr medium-entropy alloy (MEA), the new features rendered by the concentrated compositions: complex stacking fault (CSF) energy (CSFE) that is varying spatially and with processing conditions, local antiphase boundary (APB) energy, unconventional twin boundary energy and martensite phase boundary energy, and elevated lattice resistance to dislocation motion because of the rugged energy landscape. We show that the presence of multi-principal elements and their variable LCO make dislocations behave differently from conventional metals and dilute solutions, in terms of slip paths and activated nanoscale segment detrapping (NSD) processes that govern alloy strength. The multitude of (partial) dislocation behavior associated with the adjustable LCOs opens new opportunities to tailor properties, through judicious choice of processing parameters, in particular, the homogenization annealing temperature ($T_a$).

## Results

### Variable LCOs in samples processed at different temperatures.
We first demonstrate the vast range of LCO in a concentrated SS of a given composition. We chose the FCC MEA NiCoCr as a model because it is a representative of multi-principal element systems and its mechanical properties are typical of (and often

better than) other quaternary and quinary HEAs[16,28]. To enable insightful large-scale molecular dynamics (MD) simulations, which are sorely needed to provide atomistic insight about dislocation behavior and adequate statistics, but have thus far been lacking in the HEA field, we developed an empirical interatomic potential for NiCoCr alloys without consideration of spin polarization (Methods). This model is designed to capture the typical features of HEAs: multi-principal constituents (equiatomic composition) with similar atomic sizes, chemical interactions consistent with typical MEA solutions, and single-phase solution but with variable LCO. A comprehensive discussion of this new potential is presented in Supplementary Note 1 (see Supplementary Figs. 1–7 and Supplementary Tables 1–2 for property validations). We then carried out hybrid MD and Monte Carlo (MC) simulations (see Methods) to obtain equilibrium configurations after annealing at different temperatures, $T_a$.

Figure 1 shows the LCO for samples annealed at different $T_a$, with LCO measured by the pairwise multicomponent short-range-order parameter[29] (see Methods). As seen in Fig. 1a, with decreasing $T_a$, the absolute values of $\alpha^1_{Ni-Ni}$, $\alpha^1_{Ni-Co}$, $\alpha^1_{Ni-Cr}$, and $\alpha^1_{Co-Cr}$ first smoothly increase when $T_a >$ ~850 K and then dramatically rise to saturated values at sufficiently low $T_a$. Meanwhile, $\alpha^1_{Co-Co}$ and $\alpha^1_{Cr-Cr}$ are slightly negative and largely remain constant for the whole $T_a$ range. This suggests that our model NiCoCr system develops local Ni segregation and cobalt–chromium (Co–Cr) ordering with decreasing $T_a$ (see Supplementary Note 2 and Supplementary Fig. 8 for $\alpha^2$ and $\alpha^3$). Note that previous density functional theory (DFT) computations[23,30] showed more Ni–Cr pairing. This difference may be due to their different assumption of magnetic effects: the magnetic states of NiCoCr with LCO developed at finite temperatures were treated with ground-state spin-polarized calculations as if they were at 0 K[23,30], without considering spin fluctuations (both thermal and static fluctuations). In our embedded-atom method (EAM) model of the alloy, the spin polarization was not explicitly considered in a scenario that could be regarded as approximately non-magnetic. These different treatments give energy difference of only a few meV/atom for the equiatomic composition, but ~15 meV/atom for NiCoCr$_{0.5}$ (see Supplementary Note 9 and Supplementary Fig. 21). This energy difference from magnetic ordering needs to be taken into account together with that due to local chemical ordering. As a result, it is expected that calculations assuming different magnetic effects would affect the magnitude of order parameters. Nevertheless, both our EAM model and previous DFT results[23,30] capture the Co–Cr ordering, which is consistent with the equilibrium phase diagram to form Co–Cr intermetallic phase(s)[31]. Such increasing LCO with decreasing $T_a$ also implies significant deviations from the configurational entropy of an ideal solution $S_{c,ideal}$. Figure 1b shows the dependence of $S_c$ on processing temperature $T_a$, based on the cluster variation method (CVM) with pair approximation[32]. The overall trend is similar to what Gao et al.[33] reported using a similar method for other HEAs. As seen, an MEA/HEA rarely reaches $S_{c,ideal}$, ~95% at best at the highest $T_a$ (1650 K). With decreasing $T_a$, $S_c$ turns away from $S_{c,ideal}$ fairly early and loses half of its magnitude when LCO becomes obvious (compare with Fig. 1a). As such, a truly random SS is only an extreme state of MEA/HEA and difficult to reach in practice. More commonly, an MEA/HEA at a given composition possesses partial chemical order.

Figure 1c–e shows three representative atomic configurations. As seen, samples prepared at relatively high $T_a$, for example, 1350 K (Fig. 1c) and 950 K (Fig. 1d), show nanoscale Ni clusters and interconnected Co–Cr clusters with relatively random compositions and orientations. Below $T_a$ ~ 850 K, for example,

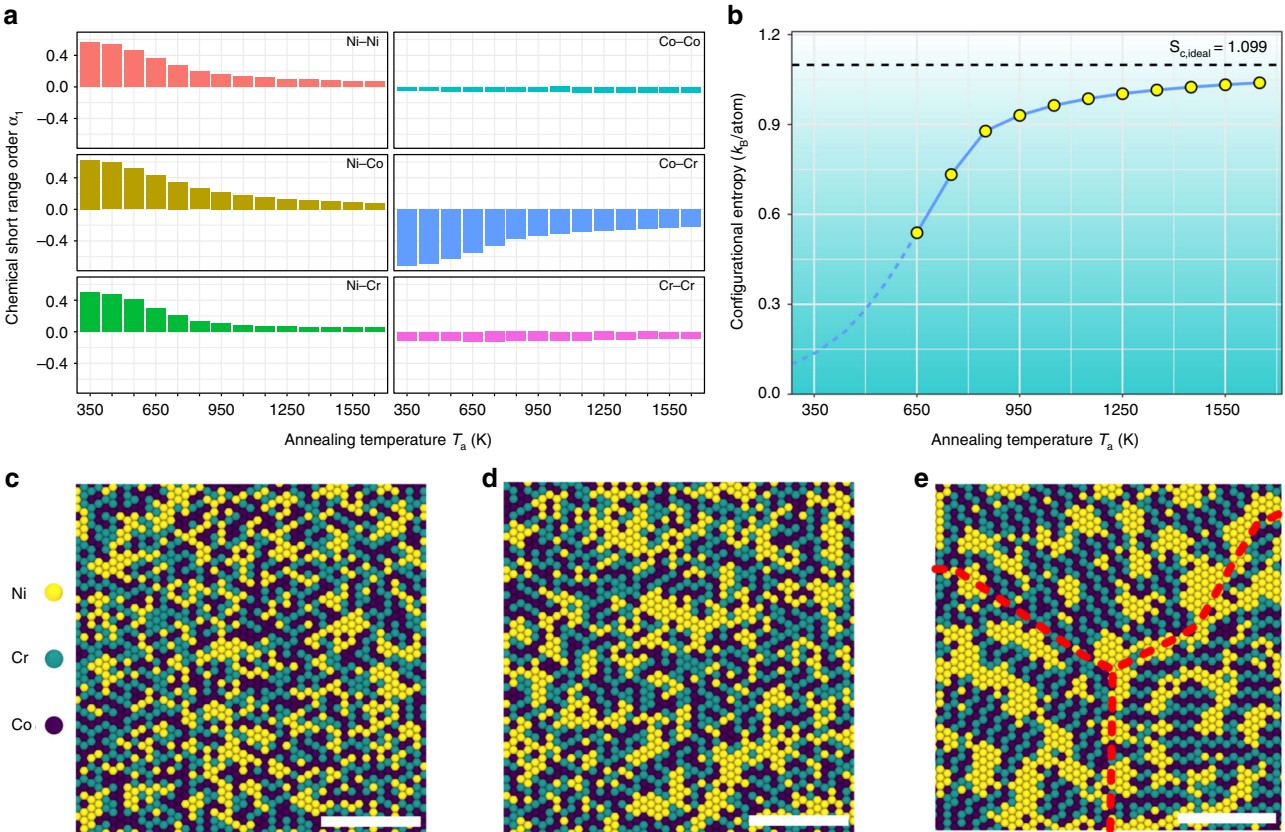

**Fig. 1** Local chemical ordering (LCO) at different annealing temperatures $T_a$. **a** Pairwise chemical short-range order parameter $\alpha_1$ (see Methods) at different annealing temperatures. **b** Configurational entropy of the NiCoCr ternary solution and its temperature dependence. Points are data estimated through the cluster variation method (CVM) with pair approximation, connected using a blue line as a guide for the eye. This approximation becomes increasingly inadequate at high LCO; thus, a dashed line is used instead to project the trend at low $T_a$. Black dashed line denotes the $S_{c,ideal}$. **c–e** Representative configurations at $T_a = 1350$, 950, and 650 K, respectively. The red dashed lines indicate the cobalt–chromium (Co–Cr) domain boundaries. The scale bar is 3 nm. All atomic configurations are viewed on the (111) plane

at $T_a = 650$ K (Fig. 1e), dramatic chemical ordering creates compositionally identical but orientationally distinguishable Co–Cr domains, as marked by the domain boundaries in Fig. 1e (dashed lines). Randomly distributed Ni nanoscale precipitates break up these Co–Cr domains into finer regions. This visually obvious LCO persists across the $T_a$ range examined here up to temperatures (e.g., $T_a = 1650$ K) near the melting point. For relatively high annealing temperatures such as those in Fig.1c, d, we expect that the kinetics needed for chemical ordering would be accessible in typical laboratory experiments: here we emphasize that local ordering/clustering or even compositional decomposition has been recently reported in several HEAs[17,34–37]. However, when $T_a$ is too low for adequate aging the predicted chemical ordering may require a timescale much longer than the typical homogenization duration in experiments. Nevertheless, kinetically permitted, all the HEAs evolve toward the ground state (even the pioneering Cantor HEA[17] is no exception). Thus, the partially ordered system is actually the norm for single-phase HEA solutions. In this context, Fig. 1e represents a microstructure at the other end opposite to the RSS. Between the two, there is ample room for structural engineering.

**Energy pathways of slip, twinning, and martensitic transformation.** We next illustrate how the dislocation behavior would change in NiCoCr MEA samples with different LCOs. Figure 2 shows the energy landscape calculated at 0 K (see Methods) for the pathways of slip, deformation twinning (DT), and FCC →

HCP (hexagonal close-packed) martensitic transformation (MT) to demonstrate the large impact of the LCOs. Figure 2a shows the energy landscape of ordinary dislocation slip in samples prepared at different $T_a$. A full dislocation slip consists of a leading partial dislocation slip (the first humps in Fig. 2a) and a trailing partial dislocation slip (the second humps in Fig. 2a). In our case, the leading partial dislocation (B → δ along the $[2\bar{1}\bar{1}]$ direction) induces both structural (FCC → HCP) and chemical changes (LCO breaking), resulting in a convoluted SFE, that is, a significant fraction of SFE is due to breaking LCOs. Such chemical contribution to SFE is expected to be ubiquitous in real-world HEAs. We therefore adopt the notation of CSF for HEAs, similar to that in superalloys. The sample-averaged CSFE varies remarkably depending on $T_a$. For example, the RSS extreme shows a negative CSFE of −24.0 mJ/m², in accord with previous DFT calculations[38–41]. The unstable SFE is also close to that reported previously[39,41,42]. However, once LCO kicks in (via annealing), the CSFE jumps to a positive range from ~1.4 to ~57 mJ/m², as $T_a$ varies from 1650 K (near the melting point) to 650 K. Despite some numerical differences due to different model assumptions and simulation procedures, such LCO-dependent SFE is consistent with previous DFT results by Ding et al.[30], where the intrinsic SFE was reported to vary from −43 to 30 mJ/m² with increasing LCO. In other words, an FCC HEA/MEA can take a CSFE value out of a wide range and does not necessarily have the low SFE anticipated[38–43]. In laboratory experiments, the measured SFE for NiCoCr[44] is ~20 mJ/m²; however, a one-to-one comparison between our model calculations and experimental

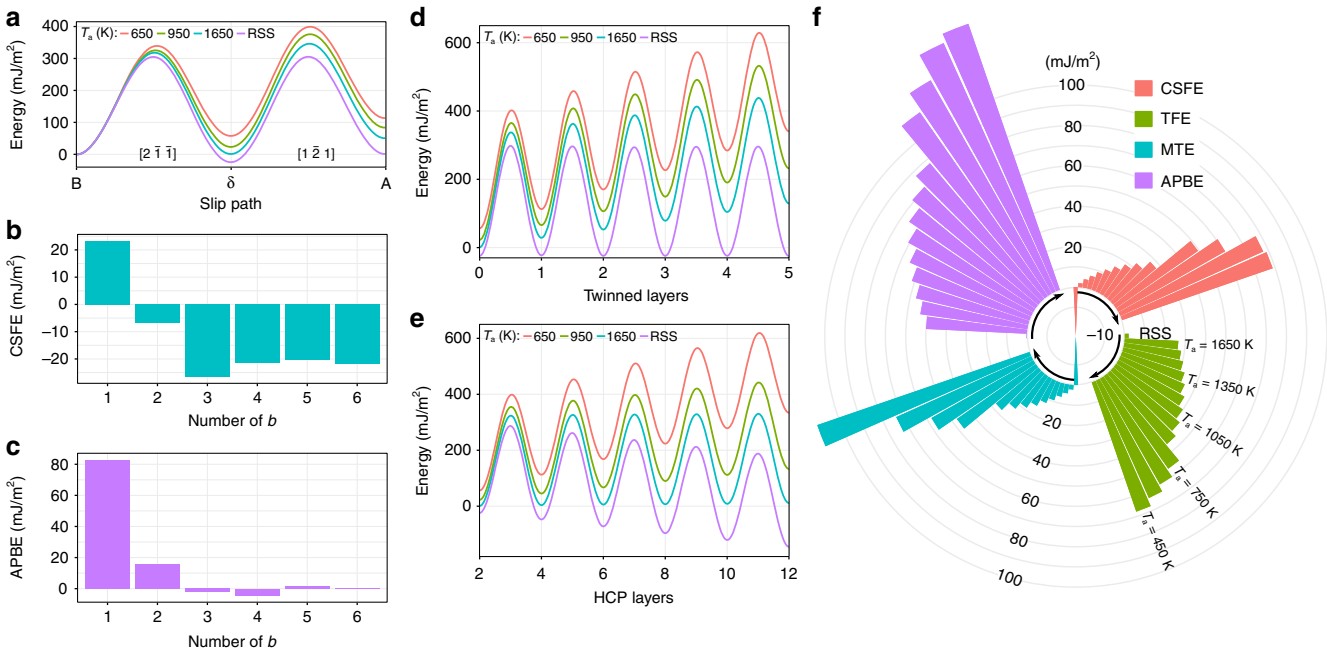

**Fig. 2** Energy landscape depends on the annealing temperature used to prepare the medium-entropy alloy (MEA). **a** Full dislocation energy pathways for the first Burgers vector. **b**, **c** Complex stacking fault energy (CSFE) and antiphase boundary energy (APBE) changes upon increasing number of Burgers vector (on the same slip plane) for a sample with $T_a = 950$ K. **d**, **e** Energy pathways for twinning and face-centered cubic (FCC) → HCP (hexagonal close-packed) martensitic transformation, respectively. Both deformation twinning (DT) and martensitic transformation (MT) are based on the CSF in **a**. **f** The average complex stacking fault energy, twin fault energy, martensitic transformation energy, and antiphase boundary energy for random solid solution (RSS) and samples with different $T_a$ (450−1650 K). The unit of the iso-energy circles is mJ/m².

results is not advisable because (a) the LCO information is unavailable for the experimental samples; (b) experiments use different method in evaluating SFE, and (c) our model merely uses an empirical potential. The trailing partial dislocation ($\delta \to A$ along the $[1\bar{2}1]$ direction) recovers the structural change (HCP → FCC) and eliminates the CSF. However, it interrupts LCO (except RSS) and creates local APBs, as indicated by the non-zero fault energies (A, Fig. 2a). Specifically, the local APB energies (APBEs) increase from ~50 to ~112 mJ/m² when $T_a$ varies from 1650 to 650 K. Such variable APBEs are expected to be common in HEAs/MEAs with noticeable LCOs[17,18,20,26,35,45,46], and play an important role in mechanical properties (more discussions later).

Due to the short-to-medium range nature of LCO, both CSFE and APBE decay with repeated slip on the same slip plane. As shown in Fig. 2b, c, beyond $3b$ slip, the average CSFE and APBE decay to values similar to the RSS. This suggests that plastic shear larger than $3b$ can destroy most of the LCO on the neighboring planes. See Supplementary Fig. 9 for similar results on other samples. However, it should be noted that dislocations would not always run on a fixed atomic plane due to extrinsic obstacles (e.g., forest dislocations), as in the case of work hardening where plastic flow spreads onto many other planes with LCO. Additionally, even if an atomic plane becomes random with repeated dislocation slip (as often involved in slip plane softening[47]), part of the atomic plane could easily gain new LCO due to the implanted extra half-plane of a cutting dislocation. In these cases, LCO obviously influences the flow stress. As a result, it takes extensive/severe plastic deformation to convert the SS into quasi-random, if that is the desired state.

DT and MT are also important routes of plastic deformation. Figure 2b, c demonstrate the energy pathways of DT and the FCC → HCP MT. Twinning dislocations glide on consecutive slip planes, while MT dislocations glide on every other plane, leading to growth steps of one layer and two layer, respectively.

Apparently, the energy pathways of both DT and MT are functions of $T_a$ and the number of transformed layers $\lambda$ (the non-zero integer in Fig. 2d, e), $\gamma(T_a, \lambda)$. The boundary energy (twin boundary or phase boundary) can then be defined as $\gamma_B = \gamma(T_a,\lambda)/2$. Again, at a given $\lambda$, samples with various LCOs produce a broad spectrum of $\gamma_B$ for both DT and MT, with RSS serving as an extreme case. For DT in RSS, both the $\gamma_B$ (~11 mJ/m²) and unstable twin fault energy (~320 mJ/m²) are similar to previous studies[39,41,42]. For MT in RSS, HCP phase growth significantly lowers $\gamma_B$, suggesting the metastable nature of the FCC RSS. However, samples with LCOs show remarkably different behaviors. For example, $\gamma_B$ in DT never converges to a constant value as $\lambda$ increases ($\gamma_B$ increases approximately in a linear fashion with respect to $\lambda$, see Supplementary Fig. 10), which is in sharp contrast to elemental FCC metals, where $\gamma_B$ generally converges to a constant value after several layers[48] such that the ensuing widening does not cost extra energy. Here for HEAs the unconventional twin boundary energies entail increasing energy penalty to break LCOs (associated with neighboring slip planes) as twin thickens. As a result, twin growth is no longer easy in HEAs, as the thickening always incurs additional energy penalty upon destroying the LCO layer by layer. Similar trend is also observed for the FCC → HCP MT. This offers an explanation to the experimental observation that separated SFs, nanotwins and very thin HCP laths dominate in deformed NiCoCr[28,42,44,49,50].

Figure 2f summarizes and compares the average values (see Methods) of CSFE, twin fault energy (TFE, i.e., $\gamma(\lambda = 1) - \gamma(\lambda = 0)$), martensitic transformation (MT) energy (MTE, i.e., $\gamma(\lambda = 4) - \gamma(\lambda = 2)$) and APBE for samples with various LCO. Here again, the RSS serves as an extreme case characterized by negative CSFE, negative MTE, and negligible TFE and APBE. For samples annealed at $T_a \geq 850$ K, the CSFE, TFE, MTE, and APBE all increase with decreasing $T_a$; however, their magnitude relative to one another remains similar to that of RSS, indicating a similarly strong tendency to form CSFs, nanotwins, and thin HCP

lamellae, consistent with the experimental observations[42,50]. For samples annealed below 850 K, all fault energies become relatively high and the energy costs for the leading and trailing partial dislocation slip become comparable, and thus narrowly extended dislocations may dominate the plastic deformation. See Supplementary Note 3 and Supplementary Fig. 11 for other LCO-dependent (or $T_a$-dependent) material properties.

**Nanoscale heterogeneities due to spatial variations of LCO.** The next important observation is that, in addition to the $T_a$-dependent average LCO of a sample, inside a specific sample there is a spatial variation of LCOs creating various nanoscale heterogeneities, leading also to wide property distributions. In other words, the dislocation behavior is spatially variable on different length scales. In Fig. 3, we plot the local CSFEs and local APBEs to show the statistics and spatial variations (see Methods). Figure 3a shows the probability density distributions of local CSFEs. As seen, for each processing condition, the local CSFEs exhibit significant spatial variations; the 25–75% accumulative probability range (highlighted by white boundaries) is generally >~ 25 mJ/m$^2$, not to mention the even wider range between the minimum and maximum values. The distributions for relatively high $T_a$ ($\geq$950K) follow a Gaussian profile, while low $T_a$ (e.g., 650 K) creates asymmetric distributions. The latter is closely related to the appreciable Co–Cr domains formed at relatively low

$T_a$, that is, each domain may have its own specific distribution (see Supplementary Note 4 and Supplementary Fig. 12 for an example) such that the merged overall distribution may no longer be Gaussian. In addition, the distributions shift to higher CSFE values with increasing LCO, consistent with the trend on the sample average in Fig. 2. Figure 3b shows an example of spatially varying CSFEs in a sample with $T_a$ = 950 K. As seen, the CSFE is highly heterogeneous over space, with many nanoscale domains showing much lower/higher values than the average. As shown in Fig. 3c, d, due to the highly localized LCOs significant spatial variations are also observed for local APBEs, which, in contrast to intermetallic compounds such as the $\gamma'$ phase in nickel super-alloy, often exhibit significant deviations from the average value.

The chemical-order heterogeneities on the nanoscale are expected to present obstacles to dislocation movement and can be exploited to enhance the strength of HEAs. In Fig. 3e–g, we show three typical dislocation configurations in samples with $T_a$ = 650, 950, and 1650 K, respectively. These dislocation configurations are the snapshots after 100 ps relaxation at 300 K and under a constant shear stress of 300 MPa (to counteract the restoring force due to the APBE). First of all, all samples show extended dislocations with apparently different average dissociation width that increases with increasing $T_a$, consistent with the trend of CSFE shown in Fig. 2. As a result, the dissociated dislocation configuration divides the crystal into three distinct regions (Fig. 3f), that is, the local APB region due to full slip, the

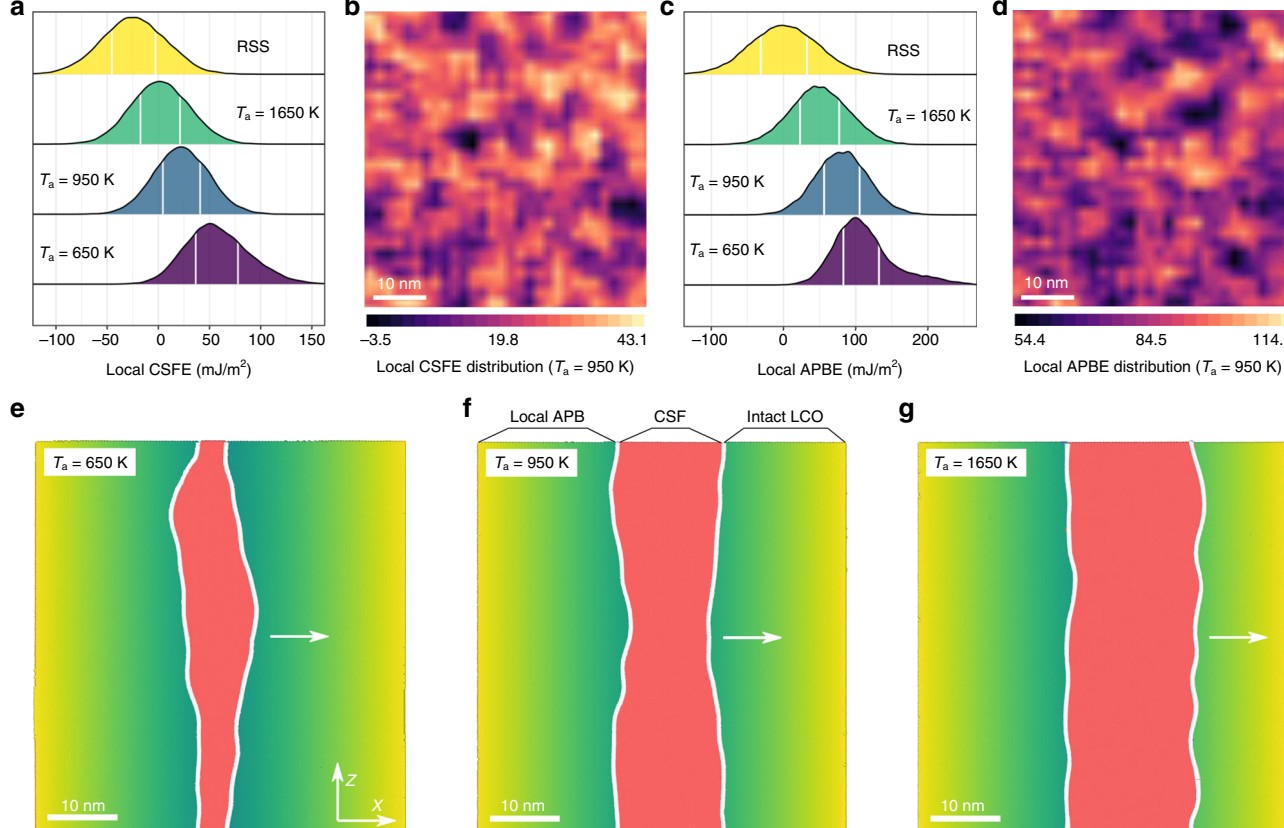

**Fig. 3** Statistical distributions of local properties and their effects on dislocation structure. **a** Probability density distributions of local complex stacking fault energies for samples with $T_a$ = 650, 950, and 1650 K, respectively, and random solid solution. The 25–75% accumulative probability range is highlighted by white boundaries. **b** Spatial distribution of local complex stacking fault energies in a sample with $T_a$ = 950 K. **c** Probability density distributions of local antiphase boundary energies for samples with $T_a$ = 650, 950, and 1650 K, respectively, and random solid solution. **d** Spatial distribution of local antiphase boundary energies in a sample with $T_a$ = 950 K. **e–g** Examples of extended screw dislocation in samples with $T_a$ = 650, 950, and 1650 K, respectively. All configurations in **e–g** are obtained after a 100 ps relaxation at $T$ = 300 K and under a constant shear stress of 300 MPa. The x- and z-direction are along [11$\bar{2}$] and [1$\bar{1}$0], respectively. Periodic boundary conditions are only applied in z-direction. Dislocation cores are represented by the white tubes and complex stacking faults (CSFs) are colored in red. Leading partial dislocation is on the right and trailing partial dislocation is on the left

CSF region due to partial dislocation slip, and the intact LCO region without slip. Second, all partial dislocations exhibit nanoscale curvatures that vary along the dislocation lines, leading to wavy dislocation lines and rugged stacking fault ribbons. These results are consistent with the spatial variations of both CSFE and APBE shown in Fig. 3a–d, that is, the dislocation line tends to bow out in soft regions, while hard regions act as obstacles that trap dislocation segments. These spatial variations of dislocation configurations clearly demonstrate that the local properties span a wide range and some extreme values may be highly relevant to the rate-limiting step of a thermal activation process[51]. LCO is the root cause of the local properties: the pronounced variations in dislocation splitting width and core structure have been reported in experimental samples[52,53] and in simulated RSS[54]. In this regard, the HEA SS is more like a cocktail of many coexisting SSs.

**Dislocation motion and LCO-induced strengthening**. Now let us take a further step to examine the motion of dislocations that carry plastic flow, and demonstrate that LCOs indeed influence how difficult it is for dislocations to move and thus the strength of the alloy. To this end, we first resolve how a dislocation actually moves in a lattice with various LCOs. Here we consider a relatively long segment of a leading partial dislocation that often controls the mobility of an extended dislocation as well as the DT or MT processes. Figure 4a shows a series of snapshots of the evolving dislocation line while it glides in the lattice, based on the MD simulation for Fig. 3f. Specifically, the leading partial dislocation is subjected to a constant shear stress of 300 MPa at 300 K. As seen, the dislocation line is wavy and does not move smoothly, due to the nanoscale LCO heterogeneities as shown in Fig. 3; instead, the dislocation moves via a series of forward slip of local segments, each is on nanoscale and detraps from its local LCO environment. Highlighted in red in Fig. 4a are the nanoscale swept areas between the start and final states (e.g., see snapshot at 20 and 60 ps, respectively) corresponding to a particular NSD event. This NSD, one at a time in an intermittent manner and one next to another locally on the dislocation line, is in sharp contrast to conventional FCC metals where dislocations move smoothly by either simultaneously propagating a long dislocation line (see Supplementary Fig. 13 for an example of Cu), or bow out in between unshearable obstacles that are spaced quite some distance (at least many nanometers) apart.

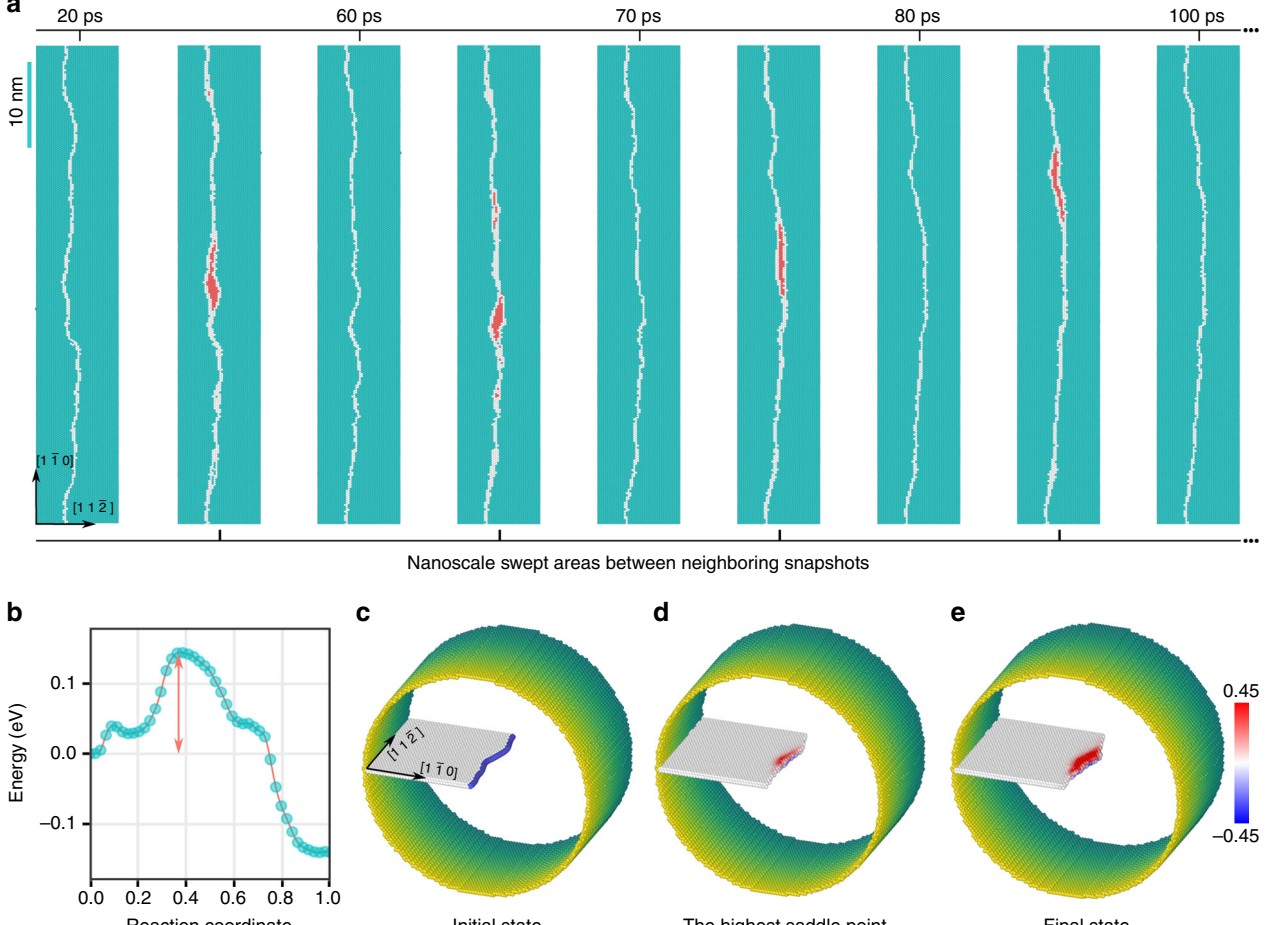

**Fig. 4** Dislocation motion via nanoscale segment detrapping mechanism. **a** Correlated nanoscale processes for a leading partial dislocation (white color) in a sample with $T_a = 950$ K. The applied temperature and shear stress are 300 K and 300 MPa, respectively. The swept areas between two neighboring snapshots/instants when the dislocation settles down briefly without motion (e.g., 20 and 60 ps) are highlighted in red. **b–e** The calculated minimum energy path of a nanoscale segment detrapping process, for a sample with $T_a = 950$ K and subject to a local shear stress of 400 MPa. See Methods for calculation details. Reaction coordinates in **b** is the scaled hyperspace arc length. **c** shows the initial configuration with a curved leading partial dislocation (the blue tube). **d** shows the configuration corresponding to the highest saddle point in **b**, and **e** shows the final configuration. For **c–e**, surface atoms are colored according to their positions along the axial direction and white atoms represent the complex stacking fault. The activated process of nanoscale segment detrapping during dislocation stick-slip motion is highlighted by the swept area as colored in red, based on the magnitude of atomic displacements along the axial direction. Numbers on the colorbar are in unit of Å

In order to evaluate the LCO effects on the barriers associated with a typical nanoscale segmented slip process, new samples with smaller dimensions (see Supplementary Fig. 14 for size effects on Peierls barrier calculation) are used to calculate the minimum energy path (MEP). Figure 4b–e shows such an example for a sample with $T_a = 950$ K (see Methods). As seen in Fig. 4b, a typical nanoscale segment movement traverses quite rugged MEP consisting of multiple finer events with variable barriers, reflecting the complex nature of the underlying energy landscape in concentrated alloys. To complete the entire process (e.g., from Fig. 4c to Fig. 4e), these finer events should be activated in a strongly correlated fashion. Here, to estimate the effective barrier associated with the entire process, we ignore those finer events whose backward barrier is smaller than the forward barrier. Then, the effective barrier is taken as the largest barrier along this modified MEP. For example, the effective barrier (marked with double-ended arrow in Fig. 4b) can be taken as the energy difference between the initial configuration (Fig. 4c) and that at the highest saddle point (Fig. 4d). Note that in elemental FCC metals, the Peierls barriers generally vanish when the applied stress is on the order of $10^1$ MPa; however, in the example shown above, the effective barrier is still ~0.15 eV even under a local shear stress of 400 MPa, indicating LCO-induced strengthening.

Next, we systematically compare the effective barriers and the resulting material strengths for samples with various degrees of LCOs. The average values (and standard deviations) of the effective barriers are shown in Fig. 5a, for samples with different LCOs and subjected to different stress levels (see Methods). See Supplementary Note 5 and Supplementary Fig. 15 for a plot encompassing all the effective activation barriers. As seen, a noteworthy trend is that a sample with stronger LCO (or processed at lower $T_a$) tends to impose larger barriers to the nanoscale segment slip process. If fed into the rate equation, these increasingly larger barriers would trap the local dislocation segment for an exponentially increasing time, thus reducing the dislocation mobility. Meanwhile, with increasing shear stresses, the average barriers for all types of samples decrease and eventually vanish at the (average) athermal stress limit, as shown by the fitted curves. Again, samples with stronger LCO show

higher (average) athermal stress limit. The complex nature of the underlying energy landscape and of the nanoscale heterogeneities are also directly reflected by the significant standard deviations for each average value. The 300 K average activation volume is shown in Fig. 5b (see Supplementary Note 6 for the evaluation of activation volume). Overall, the activation volume associated with the NSD process at normal shear stress levels is in the range of $10^1 b^3$–$10^2 b^3$, consistent with some recent experimental measurement[14,16] and the direct MD observations shown in Fig. 4a. These relatively small activation volumes suggest a thermally activated process much more sensitive to temperature and strain rate than conventional FCC metals where the activation volumes are much larger.

The intrinsic shear strength can then be evaluated by solving Orowan's equation (see Supplementary Note 6), and the results are shown in Fig. 5c for different samples at 300 K. As seen, at a given strain rate, the shear stress required to activate NSD increases considerably with increasing LCO (or decreasing $T_a$), clearly demonstrating the pronounced strengthening due to LCO. Such LCO-induced strengthening was indeed observed in experiments on a BCC HEA where the yield strength was increased by 76% after one-day annealing of the as-cast state, which was attributed to the development of short-range clustering[34]. Here the activation shear stress is taken as the intrinsic shear strength of the MEA lattice, because the mobile dislocation density was assumed to remain on the level for well-annealed samples ($10^8/m^2$, see Supplementary Note 6) and extrinsic (dislocation) interactions have not yet kicked in. Since HEAs/MEAs have been assumed to be RSS in many experimental and computational works, here we estimate the critical resolved shear strength (CRSS) of NiCoCr RSS under experimentally relevant conditions. As shown in Fig. 5c, at 300 K and $10^{-3}$ s$^{-1}$ (as marked by the red point), the CRSS is ~70 MPa. Experimentally measured CRSS is ~69 ± 3 MPa for single-crystal NiCoCr MEA[55] (axial yield strength of polycrystals can be above 270 MPa[16,28,44,56], but there some grain size strengthening is likely involved).

We also observed similar LCO-induced strengthening effects for extended edge dislocations, using MD simulations at 300 K under a constant strain rate, see Supplementary Note 7 and

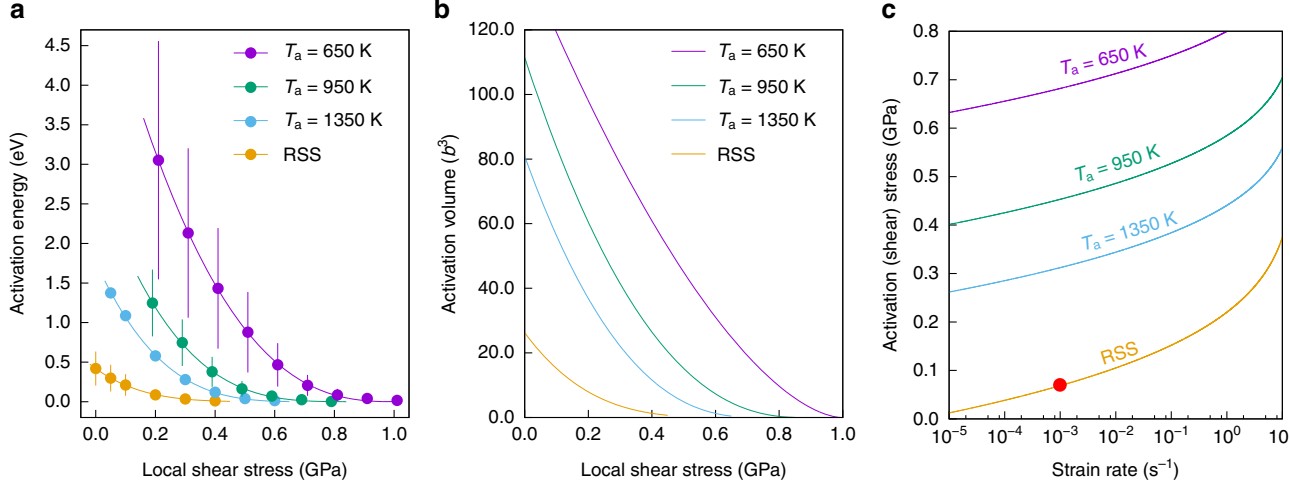

**Fig. 5** Local chemical ordering (LCO)-induced strengthening. **a** The average activation barriers associated with the nanoscale segment detrapping process. For visualization, data for $T_a = 950$ and 650 K are horizontally shifted by −0.01 and 0.01 GPa, respectively. The calculated data points are fitted using $Q(\tau) = Q_0 \left(1 - \frac{\tau}{\tau_{ath}}\right)^\alpha$, where $Q_0$, $\tau_{ath}$, and $\alpha$ are fitting parameters. Error bars correspond to the standard deviations of each data point. **b** The average activation volume at 300 K (see Supplementary Note 6 for calculation details). **c** The average activation (shear) stress at 300 K as a function of strain rate. These average activation (shear) stress can be considered as the intrinsic shear strength, which for RSS is ~70 MPa (red point) at $10^{-3}$ s$^{-1}$ (comparable with experiments). With increasing LCO (or decreasing $T_a$), the intrinsic shear strength is predicted to increase, demonstrating LCO-induced strengthening

Supplementary Figs. 16–18. The shear stress required to drive forward a leading partial dislocation needs to increase by as much as a factor of three, when an RSS has been annealed at 650 K to increase LCO. Also see Supplementary Movies 1–2 for the dynamic motion processes with dislocations evolving into rugged morphologies, detrapping from the nanoscale LCO heterogeneities.

Note that the observed strengthening effect is not due to the lattice distortions caused by atomic size mismatch, as our MD/MC aging at $T_a$ alleviates the effects of atomic size mismatch through rearranging unfavorable local atomic environments. Two mechanisms are believed to be responsible for the additional strengthening due to LCO. First, the spatially heterogeneous complex SFE and APB energy results in extra restoring forces on a moving dislocation that breaks LCOs. The stronger the LCO, the larger restoring force a dislocation feels on average. Second, the spatially varying LCO presents nanoscale heterogeneities acting as trapping roadblocks to moving dislocations, in a way similar to G-P zones. In comparison, in RSS without a dominant LCO, the dislocation line still shows obvious waviness and its forward motion would still undergo NSD (see Fig. 6), such that the RSS is already strengthened compared to elemental FCC metals. However, in the random solution the NSD only needs to detrap from local favorable environment formed due to statistical fluctuation, such as spatial fluctuations deviating from the nominal full chemical disorder (see Fig. 6). Such strengthening behavior is akin to the Labusch theory[57] where strengthening is attributed to the collective interactions between many solute atoms and a dislocation. In other words, it is the favored statistical fluctuations in solute configuration that locally pin or at least trouble dislocations, rather than individual solute atoms. When a dominant LCO emerges in a sample, the activation barriers for local dislocation segments would further increase as shown in Fig. 5, the dislocation lines become increasingly wavy due to partially ordered spatial heterogeneities, and the critical resolved shear stress rises further. Such analysis provides an atomistic explanation for the experimentally observed elevation of Peierls stress in HEAs/MEAs, for example, the lattice friction stress of NiCoCr was found several times higher than that of elemental FCC metals;[56] and the yield strength of TaNbHfZr was increased by up to 76% via annealing-induced ordering at various intermediate temperatures[34]. As a practical approach to take advantage of this LCO strengthening, one can rapidly cool an HEA to obtain an RSS to make use of its room temperature ductility for shaping, and afterwards age it at an elevated temperature to acquire adequate LCOs to raise its strength for service (e.g., when high strength is needed for use at elevated temperatures).

## Discussion

To recapitulate, our atomistic modeling and simulation, using a non-magnetic ternary NiCoCr MEA as the model system, suggest that generally the multi-principal-element alloys are not ideal or regular SSs, and do not have a high configurational entropy close to random mixing as depicted in the literature. The most probable structural state contains partial chemical order. The configurational entropy is reduced from ideal solutions, even for HEAs/MEAs prepared at rather high temperatures. They are nevertheless still of high entropy compared to the ground-state

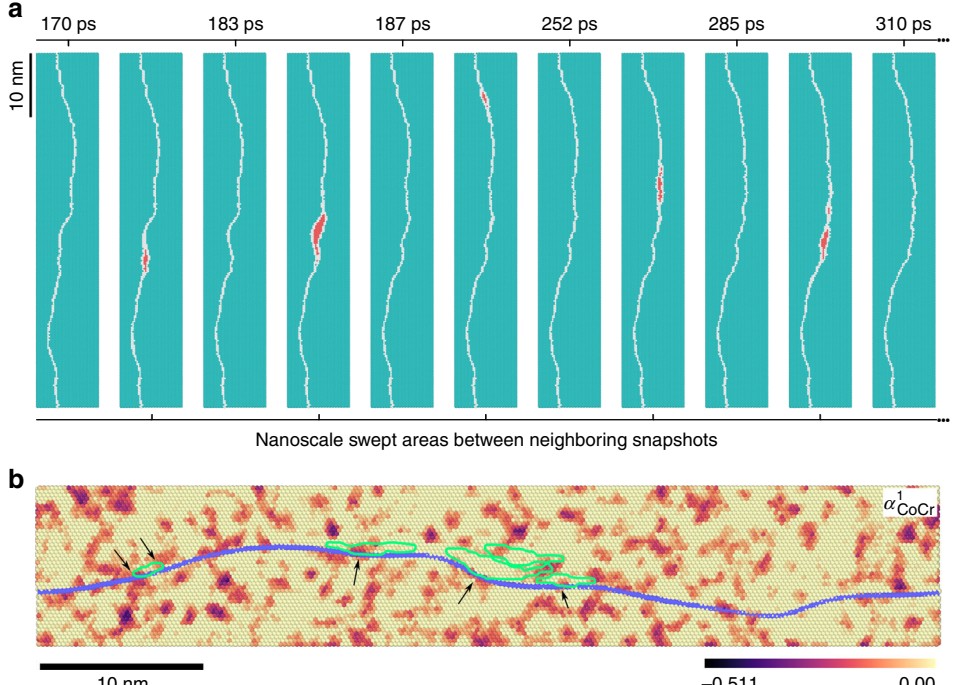

**Fig. 6** Nanoscale segment detrapping in random solid solution of NiCoCr. **a** Dislocation line morphology evolution under a shear stress of 50 MPa at 300 K. The several swept areas are highlighted in red. **b** Correlation between nanoscale segment detrapping events and spatial deviations from chemical disorder. Atoms are colored according to the coarse-grained $\alpha^1_{CoCr}$; darker color means stronger Cobalt-Chromium (Co-Cr) short-range-order (more negative $\alpha^1_{CoCr}$). The swept areas in **a** are now outlined in green. These activated nanoscale segments are seen to detrap from some of the local hard regions that have stronger Co–Cr local chemical ordering (LCO) (as denoted by the arrows) and then propagate into local regions without such chemical order. It is not expected that our specific loading conditions within the short simulation period would be able to activate all the segments at LCO trapping sites. For some of the regions with weak Co–Cr LCO along the dislocation line, thermal activation is expected to have already moved those segments, producing a wavy dislocation line before loading, when the sample was first relaxed at 300 K for 100 ps. Continued bow-out increasing curvatures is limited by the line tension

traditional SSs and intermetallic compounds, arising from the variability and vastness in the internal LCO configurations when multiple chemical species are alloyed in concentrated proportions.

The significance lies in the realization that different from traditional SSs where the (dilute) solutes are approximately random, delivering almost invariable properties independent of processing, here the variable chemical order can be selected through judicious alloy processing, opening up rich possibilities, including ordering vs. segregation, degree of LCO, species involved in ordering, length scale reached through the first, second, and third nearest-neighbor layers to even nanometer domains, and spatial variation across the sample, all in a single-phase SS before any second phase precipitates out, even though the LCOs are difficult to resolve and quantify in diffraction experiments.

A specific example we have focused on is that the concentrated SS with LCOs opens a new playground for dislocations, including important benefits. First, the variable and partial chemical order explain on atomic scale the multitude of possible dislocation mechanisms: we have quantitatively mapped out the complex energy landscape, as well as the activation barriers on the rugged kinetic pathways, from the standpoint of the moving dislocation. A consequence is that, depending on how the sample is prepared, the preferred deformation option can be one, or a subset, on the selection menu, including extended dislocations, stacking faults, deformation twining, or MT. Second, the variability above explains why macroscopically measured mechanical behavior such as strength, strain hardening, and ductility can seem inconsistent at a given overall composition, when the same alloy is processed differently to reach different levels of sample-average LCOs. Third, the LCO variety can explain the observation of spatial property variations from one local region to another, in a given sample. An example is the dislocation splitting width and local SFE reported in experiments[53]. Fourth, the added dimension of LCO control offers a new knob to turn, to enable unprecedented property tuning in a single-phase solution alloy, such as that demonstrated in ref. [34]. The LCO also gets reduced towards RSS upon consecutive dislocation slips during extensive plastic deformation.

These features are expected to be general for other MEAs/HEAs where complex chemical interactions are involved. We emphasize that this flexibility afforded by MEAs/HEAs is inaccessible by its counterpart extremes: on the one end is the singular chemically disordered RSS, and the other is the ground-state phases (terminal SSs and intermetallics that are already commonly known), both offering limited property choices. In general, existing engineering alloys typically contain some ordered phases; now in concentrated single-phase MEAs/HEAs their equivalent is the LCOs. This hinges on, of course, that LCOs can bring drastic changes to dislocation properties, which has indeed been demonstrated explicitly and quantitatively in our discussions above in connection with related experimental observations. Specifically, MEAs/HEAs show spatially heterogeneous CSFs that are neither the simple intrinsic stacking fault associated with RSS (either concentrated or dilute) nor the long-range ordered CSFs in intermetallics (superlattices). MEAs/HEAs demonstrate spatially varying APBE, unlike RSS (no APBE) or intermetallics (spatially uniform APBE). Furthermore, for a specific material property while an RSS (or intermetallic) always gives a unique value, an MEA/HEA can be made to access different values, depending on how the sample is processed. In addition, DT and MT in MEAs/HEAs require significantly higher energy input than in random SS, as a result of breaking LCOs layer by layer. Finally, MEAs/HEAs with increasing LCOs are expected to exhibit strengthening, because the moving dislocations now have to undergo NSD to traverse local heterogeneities that impose

escalating activation barriers with increasing LCO. These rich structural features and dislocation energetics/dynamics answer the fundamental material science question as to what is new to the MEAs/HEAs to distinguish them from the terminal SS and intermetallic compounds. In sum, the emerging MEAs/HEAs open opportunities for structure–property design through processing to tune LCOs: "there is plenty of room at the bottom" above and beyond what is available with traditional SS alloys.

## Methods

**Empirical interatomic potential development**. An empirical potential for non-magnetic NiCoCr has been developed in the formalism of EAM[58–60], by matching a large ab initio database established for the ternary system without explicit consideration of spin polarization. The ab initio database includes a large set of atomic configurations with corresponding cohesive energies, atomic forces, and stress tensors. A similar force-matching method has previously been employed to develop a highly optimized potential for the Zr-Cu-Al system[60]. In this work, special attention has been paid to the energetics of the stacking faults and chemical ordering of Ni–Co–Cr SSs.

In order to give an accurate account of the Ni–Co–Cr system in the full compositional range, more than 3000 atomic configurations were selected to build a comprehensive ab initio database from non-spin-polarized DFT calculations. The atomic configurations not only encompass all the intermetallic compounds reported in the Ni–Co–Cr system but also include liquid/glass structures, various types of defects, transition pathways, and so on, covered in a large pressure–temperature phase space of Ni–Co–Cr. The revised Potfit code[59] was used for potential fitting. Lastly, the potential was improved through an iterative process and was further refined to match experimental data, including cohesive energies, lattice parameters, elastic constants, and phonon frequencies of the constituent elements. For more details of constructing the database and potential fitting methodology, the readers are referred to refs. [60,61].

All ab initio calculations were performed with the density functional-based Vienna Ab-initio Simulation Package (VASP)[62]. We used the projector augmented-wave method[63,64] to describe the electron–ion interactions and the generalized gradient approximation for exchange-correlation functionals. The valence electrons of Ni, Co, and Cr were specified as $3d^84s^2$, $3d^74s^2$, and $3d^54s^2$, respectively. The spin-polarization effect was not considered in the potential fitting (see Supplementary Note 9 for a discussion on magnetic effects). For high-precision ab initio total-energy calculations, we typically used $3 \times 3 \times 3$ Monkhorst-Pack[65] k-point grids with each atomic configuration containing 100–200 atoms.

**MD and MC simulation**. The chemical potential differences with regard to Ni were determined by hybrid MD and MC simulations under the semi-grand canonical ensemble at 1500 K. The set of parameters that minimizes the composition errors with respect to the equiatomic concentration are: $\Delta\mu_{\text{Ni–Co}} = 0.021$ eV and $\Delta\mu_{\text{Ni–Cr}} = -0.31$ eV. This ensures the miscibility of all elements near the equiatomic compositions. Then, hybrid MD and MC simulations, under the variance-constrained semi-grand-canonical ensemble[66], were carried out to obtain the equilibrium configurations at different annealing temperatures $T_a$. The variance parameter $\kappa$ used in our simulations is $10^3$. Our samples consist of $N = 1,584,000$ atomic sites with x, y, and z along the $[11\bar{2}]$, $[111]$, and $[1\bar{1}0]$ crystal directions, respectively. The dimension of the simulation box are ~52 nm × 6 nm × 52 nm. Periodic boundary conditions were applied in all directions. For every 20 MD steps, there is 1 MC cycle, which consists of N/4 trial moves. The MD timestep was set to 2.5 fs. A Nose–Hover thermostat and a Parrinello–Rahmann barostat were used to control temperature and pressure, respectively. Sufficient MC cycles (270,000–500,000 cycles depending on annealing temperature) were carried out to achieve converged LCO. The converged configurations were then quenched to zero temperature and energy minimized to eliminate thermal uncertainties for subsequent property calculations (e.g., LCO and various fault energies). All simulations were carried out using the LAMMPS package[67] and the atomic configurations were visualized with the Ovito package[68].

**Chemical short-range-order parameters**. The pairwise multicomponent short-range order parameter is defined as $\alpha_{ij}^m = \left(p_{ij}^m - C_j\right) / \left(\delta_{ij} - C_j\right)$, where $m$ means the $m$th nearest-neighbor shell of the central atom $i$, $p_{ij}^m$ is the average probability of finding a $j$-type atom around an $i$-type atom in the $m$th shell, $C_j$ is the average concentration of $j$-type atom in the system, and $\delta_{ij}$ is the Kronecker delta function. For pairs of the same species (i.e., $i = j$), a positive $\alpha_{ij}^m$ suggests the tendency of segregation in the $m$th shell and a negative $\alpha_{ij}^m$ means the opposite. In contrast, for pairs of different elements (i.e., $i \neq j$), a negative $\alpha_{ij}^m$ suggests the tendency of $j$-type clustering in the $m$th shell of an $i$-type atom, while a positive $\alpha_{ij}^m$ means the opposite. For a specific $T_a$, $\alpha_{ij}^m$ are the average values over a series of converged/equilibrium configurations.

**Energy pathway calculations**. All energy pathways were calculated using the equilibrium configurations obtained from the hybrid MD and MC simulation. The as-prepared samples have 30 (111) layers in the y-direction; however, we doubled the y-direction box length by replicating the sample along [1 1 1] direction. Then two neighboring (111) planes were chosen and centered in the simulation box. The simulation box was then divided into two slabs by a cutting plane dividing the two chosen (111) planes. Afterwards, the boundary conditions along [1 1 1] was switched to free surface. CSFs were created by relatively displacing the two slabs according to the Burgers vector of partial dislocations in all three <112> directions. Local APBs can then be created by relatively displacing the two slabs according to the appropriated Burgers vectors of trailing partial dislocations. For DT and MT, samples with a CSF were used as the starting configurations. DT nucleation and twin thickening were realized by repeated partial dislocation slipping on consecutive (111) planes, while MT was carried out by repeated partial dislocation slip on every other (111) plane. Energy pathways were obtained by constrained energy minimizations on intermediate configurations (only allowed to relax along the [1 1 1] direction) linearly interpolated between the initial and the final configurations (both initial and final configurations were fully relaxed). For each $T_a$, the sample-average values of various fault energies were computed based on 90 different energy pathways (30 layers each with three different Burgers vectors). For calculations on the local complex SFEs and local APB energies, we divided the whole sample into columns along the [1 1 1] direction and each column shares the same cross-sectional area of 3.2 nm$^2$. Then, the local SFEs are calculated by considering the potential energy changes of these columns. This method leads to an average value same as the global value. For better visualization in Fig. 3b, d, linear interpolation is carried based on the coarse-grained value (over the first nearest neighbors) for each local area.

**Activation barriers of NSD events**. In order to examine the LCO effects on local dislocation segment activation barriers, hybrid MD and MC simulations were performed on relatively smaller samples to obtain equilibrium LCOs at $T_a = 650$, 950, and 1350 K, respectively. For meaningful statistics, 30 different samples were obtained for each processing condition. The dimensions for these samples are 20 nm × 20 nm × 10 nm, along $x[1\,\bar{1}\,0]$, $y[\bar{1}\,\bar{1}\,1]$, and $z[1\,1\,\bar{2}]$, respectively. Similarly, after the MD/MC simulations, samples were quenched to 0 K and energy minimized. Then, cylinder configurations with axial direction along $[1\,1\,\bar{2}]$ and a diameter of 20 nm were cut from the fully relaxed bulk samples. For each cylinder sample, a partial dislocation with screw character (Burgers vector is along $[1\,1\,\bar{2}]$) was introduced at the center, based on the anisotropic elasticity theory. The dislocation line length (i.e., the axial length of the cylinder sample) is 10 nm, as the observed detrapping processes are on nanoscale. The outer layers of atoms with a thickness larger than two times the potential cutoff radius were fixed during the subsequent energy minimization. An inner cylindrical region with a diameter of 10 nm was used to calculate the average local shear stress. To create a final state for the MEP calculations, quasi-static athermal shearing (i.e., a small uniform shear strain + energy minimization at each step) was employed to trigger events at the athermal stress limit. Then, the configuration with new events was unloaded to desired strain levels, serving as the final states. MEPs were then calculated using the simplified and improved string method[69]. FIRE algorithm[70] was used to update each image during the evolution step and linear interpolation was used for the parameterization. Reparameterization was carried out every 10 increments of the minimizer. We stop the iteration once the displacement of every image between two consecutive iterations is <10$^{-3}$ Å or after a total number of 3000 iterations. We have double checked that the calculated MEPs are comparable to well-converged nudged elastic band calculations (see Supplementary Note 8 and Supplementary Figs. 19 and 20 for verification).

## Data availability

All relevant data are available from the authors, and/or are included within the manuscript and Supplementary Information.

## Code availability

All relevant codes are available from the authors upon request.

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

## Acknowledgements

This work was supported at JHU by NSF-DMR-1804320. Q.J.L. and E.M. acknowledge the computational resources at Maryland Advanced Research Computing Center (MARCC). Part of the computations was also supported by the Texas Advanced Computing Center (TACC) at The University of Texas at Austin. The work at GMU was supported by the NSF under Grant No. DMR-1611064. We also thank W. Curtin and W.G. Nöhring for helpful feedback during the EAM potential development.

## Author contributions

All authors contributed to the design of the project, data analysis and the discussions. Q.-J.L. carried out the MD simulations. H.S. developed the interatomic potentials. Q.-J.L. and E.M. led the writing of the paper.

## Additional information

**Competing interests:** The authors declare no competing interests.

