## [Peer Review File · Nature Communications]

Reviewers' comments:

Reviewer #1 (Remarks to the Author):

This paper presents an atomistic study of solid-solution strengthening due to local chemical ordering (LCO) in a medium-entropy alloy of NiCoCr. The authors developed an EAM potential for the alloy and used this potential to investigate the impact of annealing temperature on LCO. They further studied effects of the LCO on planar fault energies as well as activation energies for dislocation detrapping.

Overall, the paper is topical and important. The modeling work has been carefully performed and described. The results are very rich, and they provide new insights into the relationship between chemical structures and dislocation activities in medium/high-entropy alloys.

The paper can be recommended for publication in Nature Communications, provided the following comments are addressed adequately.

Fig. 4b-e shows the reaction pathway calculation for a leading partial dislocation in a cylindrical cell with the axial direction along $\langle 112 \rangle$. This setup is also used to calculate the energy barriers shown in Fig. 5. However, the orientation of this simulation cell is different from that in Fig. 3 and Fig. 4a. To avoid possible confusion, the crystallographic orientations (e.g., along the axial and dislocation glide directions in Fig. 4c) should be clearly labelled.

It is not clear why the authors employed the string method, as opposed to the commonly-used nudged elastic band (NEB) method, to calculate energy barriers for dislocation detrapping. For the minimum energy path shown in Fig. 4b-e, what is the corresponding NEB result? It would be helpful to use the NEB method for verifying this result from the string method. In particular, the convergence criteria (i.e., image displacement and maximum number of iterations) used by the authors may not give a truly converged result, when the transition pathway is relaxed on a slowly changing landscape. In addition, given the rugged energy landscape governing the dislocation detrapping response in complex alloys, there are possibly multiple transition pathways between the same set of initial and final states. How did the authors deal with this kind of situation?

The MD results of edge dislocations in the Supporting Information are very interesting. However, the simulation setup is not clearly described. What is the simulation geometry? Did the authors prescribe velocity at the boundary layer of the simulation cell? A schematic illustration should be helpful. These details should be provided, as they may help understand why the applied load for

detrapping a trailing partial nearly doubles that for a leading partial for Ta of 950K and 1350K. These results are likely caused by the displacement-controlled mode of loading and possibly affected by small in-plane sizes – in either x or y direction or both.

The Labusch theory has been widely used to understand the solid-solution strengthening effects in medium/high-entropy alloys. The trough model developed by Kocks and others has been often used to explain the temperature and strain-rate effects on solid-solution strengthening. What are the implications of the present work on the application of these classical theories to medium/high-entropy alloys?

Line 105, the authors state “This tendency is consistent with the equilibrium phase diagram to form Co-Cr intermetallic phase(s)”. Please provide a reference for this statement.

Reviewer #2 (Remarks to the Author):

Li et al systematically studied the local-chemical-order effect on the deformation behavior in NiCoCr medium-entropy alloys by largescale atomistic simulation. They developed a realistic interatomic potential for the NiCoCr system and demonstrated that the local chemical order can produce a wide range of planar fault energy, heighten the ruggedness of the energy landscape and raise activation barrier governing dislocation activities.

This manuscript is very nicely written and the reading is delightful. The atomistic simulation is pretty solid and extensive data have been provided (especially in SI). The following are my comments:

1) The authors insisted they developed the realistic interatomic potential for the NiCoCr systems (with extensive validation in SI). But as mentioned in the Methods, the spin-polarization effect was not considered in the potential fitting. That adds concerns to such empirical potential. I think the effect of spin-polarization is nontrivial, considering the large difference of lattice size between the non-spin-polarized NiCoCr (see Ref. 36) and spin-polarized NiCoCr (see Ref. 35,39,40). Also, the calculated intrinsic stacking fault energy are different with or without considering magnetism (see above reference). More importantly, I think the spin-polarization effect becomes more pronounced for their hybrid MC/MD simulation. That's because the local magnetic moment can be varied after each MC swaps and the accrual effect would be nontrivial after millions of swaps. I understand it's

not easy to develop a highly optimized potential for multicomponent alloys, especially for those with magnetism. I would suggest looking further into those concerns.

2) As shown in Fig. 1, they indicated the aggregated Ni atoms in the nanoscale domains. This looks very interesting. But I also find other previous literature (Ref. 23 and 40) using DFT calculation revealed the different short-range order after MC simulation. I know the sample in DFT is indeed much smaller than that in present work and the MC steps is also very limited. Could the authors have a test to use their potential and MC methods in a small configuration with limited MC steps? This can help the validation of present work. Also, Ref. 34 has revealed the nanoscale clustering of oxygen in a BCC HEA by atom probe tomography. So if there is indeed the domains of aggregated Ni atoms as predicted in present work, the direct experimental characterization would provide very strong evidence.

In sum, I can't recommend this manuscript for the publication at Nature Communications. I would reconsider it if the authors can solve my questions and concerns.

Reviewer #3 (Remarks to the Author):

Report to NC

I have seen this work two times when it was submitted and resubmitted to Nature Materials. It is sad that the authors consider alternative journals/publicity instead of properly revising their work. I see very little changes here compared to the previous versions and therefore I am very sorry but I cannot be more positive about it for Nature Communications either.

With reference to my second report to NM:

1) The authors refer to Niu, C., et al. Nature Comm. 9, 1363 (2018) when presenting their additional spin fluctuation study. To the present referee this is very unclear and it would be more appropriate to give details about those additional tools, models and calculations and explain why no thermal spin fluctuations were accounted for here.

2) The experimental SFE is quoted in the present work as well. No comments on its meaning and errors are given. Instead they associate the large (about 20 mJ/m²) experimental value to LCO. Might be very misleading.

In addition, see my previous reports to the NM version of the present manuscript.

In short, the present work is a valuable attempt to revise the common picture behind high entropy in high entropy alloys. It is demonstrated on a model system without aiming for completeness and without direct verification. It is not considered for truly high-entropy cases where entropy could easily overcome the predicted LCO versus annealing temperature behavior. Although I would like to see this work published in some way but a substantial restructuring, reformulation and softening of the conclusions are needed to avoid confusion and misinterpretations.

First report to NM

The authors discuss the impact of (tunable) local chemical order (LCO) on the properties of multi-principal element alloys. They chose a model system, the ternary NiCoCr alloy, to illustrate the LCO versus annealing temperature and the consequences in terms of dislocation mechanism and strength. The overall message of the manuscript is that local chemical order (of the size of dislocation nucleation) strongly affects the behavior of the multicomponent systems which can be captured neither by random solid solution models nor by ordered intermetallics. Despite of the importance of the disclosed phenomenon, the manuscript remain on very technical level, with several flaws and unclear details, approximations and exaggerated statements which make me suggest substantial revision and submission to more specific journals.

The title is about high-entropy and the text also refers at many places to the so-called high-entropy alloys. The entire manuscript is about a ternary model system (non-magnetic NiCoCr) which is not even an appropriate model for the previously studied NiCoCr alloy. In that respect, this work should be considered as a demonstration of a particular (although important) mechanism of chemical ordering using a model system. Whether the same conclusions hold for real high-entropy alloys remains a question to be considered in the future. This does not decrease the impact and usefulness of the present work instead places it on a specific track which the authors should realize and accept.

Identifying the vastness of LCOs, their impact on the mechanical properties and the fact that LCOs strongly depend with alloy processing is the true value of this work. It can explain the richness of properties researchers report in various publications (sometimes rather different parameters for the same system). The authors demonstrate the role of LCOs using important mechanisms of plastic deformation. The reader might wonder how other properties such as the elastic parameters, phonons or formation energies are affected by LCOs. The authors might consider adding such discussion to their work.

The authors write that the entropy is high not because of randomness but because of the “vastness of possible LCO configurations”. In other words, the large number of different local chemistry gives the entropy. Although the two concepts are different, they lead to the same overall random behavior (depending on the characteristic size of LCO, which has to remain on atomic scale to ensure the “vastness”).

Interatomic potential for NiCoCr: the authors establish the potential by neglecting the magnetism degree of freedom. This is probably OK as long as the potential capture (implicitly) the magnetic contributions to various parameters entering the fit. Then they use this “non-magnetic” interaction to describe various LCO. What about the magnetic state of certain LCO? It is not difficult to imagine that local chemistry alters the magnetic behavior (since all three components are magnetic). For instance, in Fig. 1d, the yellow area (Ni-rich) is likely to have totally different magnetic state compared to the rest of the alloy. This effect if not accounted for in the present work and the possible consequences are not discussed.

In connection with Fig. 2, they say “HEA rarely reaches the high S...”. However, the present model system in the accepted terminology does not classify HEA (perhaps medium-entropy alloy). The acronym MEA is not defined.

In the following part (page 6), they refer to “Thompson’s tetrahedron”. Either explain it for the non-familiar reader or simply describe the deformation without referring to such professional terminology.

The planar fault energies on Fig. 3 represent the average values obtained for supercells consisting of $N = 1584000$ atoms, and having 30 (111) layers in the z direction. In the method section, they say that “At each T_a , the sample average values of various fault energies were computed based on 30 different energy pathways.” What about the different stacking fault directions within one specific plane? Due to the low symmetry the three slip directions are not equivalent so in principle they should have 3×3 different planar faults. More discussion is needed to clarify the details.

They reproduce the configurational average value of the ISFE for random alloy as reported e.g. by Zhang et al, Nat. Commun. 8, 14390 (2017). With increasing degree of ordering (which is driven primarily by local Ni segregation), the ISFE increases towards the large value of Ni. One should notice that non-magnetic or ferromagnetic Ni has large ISFE, pure Co has negative ISFE at 350 K, whereas Cr prefers cubic lattice so it is very likely to have negative ISFE. In that respect, the chemical order-increasing trend of ISFE is not surprising assuming such degree of ordering can indeed be realized in the real alloy. Some comments of the observed trends would be welcome.

From what is said, one understands that the first slip in Fig. 3a (A-delta) is the stacking fault formation whereas the second slip (delta-B) is the full slip (having 60 degree different direction compared to the first part). Giving the actual slip directions would be useful for the reader. Then the first maximum is the so called unstable stacking fault energy and from the figure one can read about 300-350 mJ/m² depending on the degree of order. The random value seems to be substantially larger than the ab initio value reported by Zhang et al. in the above Nat. Commun. publication (264

mJ/m²). Should that be considered as the accuracy of the EAM for the generalized stacking fault energy surface or is the difference in the simulation boxes? On the other hand the present ISFE agrees surprisingly well with the above ab initio result (-24 mJ/m²). Another recent publication by Huang et al. Nat. Commun. 9, 2382 (2018) reports about 300 mJ/m² for the unstable stacking fault energy of the completely disordered NiCoCr. More comments are needed to compare the present predictions for random cases with previously reported values and thus verify the validity of the EAM for the present problem.

The present APB are due to the LCO. Dislocations destroy the LCO and thus lead to APB (which is a rather large energy especially at temperatures where substantial LCO exists). This also means that high dislocation activity reduces the impact of LCO bringing the alloy back to the quasi random behavior. This is to some extent confirmed by Fig. 3b and nicely discussed in the main text. The unstable twin fault energy should also be contrasted with the ab initio value reported in the above reference.

On page 10 the special variations of the planar fault energies is presented. The local fault energies correspond to 3.2 nm² areas. No details are provided how the local excess energy is defined (which the end should give on the average the global energies presented in Fig. 3).

Fig. 4b shows the distribution of the local fault energies. Even at low temperatures, the distribution includes negative ISFE regions. This should represent sources for martensitic transformations. However, no such transformation was observed in recent tests by Gludovatz et al. Nat. Commun. 7, 10602 (2016). Could that be due to the extremely low temperature used in those tests? (Unfortunately the present work does not address temperatures below the room temperature). If yes, how can one understand the observed nano-twinning when the ISFE is so large (as suggested by the present work)?

Fig. 4c is presented also in the supplementary material (Fig. S5). Please explain the reason.

Fig. S3 compares the ab initio and EAM results for the equation of state. Although not explicitly mentioned, I assume the ab initio is also non-magnetic here. What about the magnetic transitions due to lattice expansion?

Fig. S4 is not discussed at all (referred to in the main text as group of figures). A similar map of ab initio should be provided and compared to the EAM results to confirm the validity of EAM for elastic and other properties. The question is the degree of LCO for these figures. Was the calculation done for completely disordered or partially ordered samples? Please explain and discuss the figures.

Fig. S5 is not discussed as all (similar to Fig. S4). The displacement is in the stacking fault direction but no unit is given. It may be assumed that ISFE is the intrinsic stacking fault position (partial Burgers vector). But the large energy barrier before (over 1 J/m²) seems totally unrealistic. What is the small barrier after ISFE? Please explain (also in the main text, Fig. 4c). It is understood from the text above Fig. S5 that here authors used random configurations (more than 10000 different configurations). Why not specific LCO configurations? Furthermore, they write “The negative stacking fault energies in the NiCoCr alloy has previously been reported”. But in the present context there is no such as a single-value stacking fault energy, only a distribution of ISFE. It is hard to follow the information they try to deliver here.

In Fig. S7 they compare the ab initio and EAM cohesive energies for various degree of orders (expressed in terms of temperature). The agreement is reasonable although the differences between the two sets of data sometimes are larger than the ordering-induced changes in the EAM energies. Another questions is the kinetics: it is unclear whether such degree of LCO is indeed permitted by the kinetics. More discussion would be needed to convince the reader.

In summary, the present work is a comprehensive attempt to discuss the impact of local ordering on the mechanical properties of a ternary model alloy. The results are interesting and perhaps also important for the understanding of the behavior of multi-principal elements alloys such as high-entropy alloys. However, the manuscript some places is not well presented, the modeling involves a large degree of uncertainty and approximations, the practical realization of the partially ordered systems is unclear, the effect of magnetism is not discussed which might become crucial in systems with substantial LCO, the interplay between LCO and dislocation activity beyond a critical strain level is not clear (perhaps LCO is important only in as-prepared samples), etc. All these issues make me to decline the present work especially for a high-impact journal. A properly revised, complemented and improved manuscript might be considered in a more specialized journal.

Second report to NM

The authors have carefully considered the points raised by the two referees. They amended the manuscript at several places reaching a more readable and perhaps also more complete work. Nevertheless, the revision does not answer several key points. Here I list some:

1) Magnetism in Ni-Co-Cr system.

They claim “atomistic simulations using realistic potentials”, “we have spent considerable time and efforts developing a realistic EAM potential for the Ni-Co-Cr system”, and at the same time admit “We agree that our simulated NiCoCr is only a model system and have added a note to that effect following the reviewer’s comment. However, we also note that our specific case is representative”, “it is not imperative for the NiCoCr model to be a 100% accurate match for real-world NiCoCr in the

lab". These statements are contradictory. Furthermore, considering that magnetism has been detected in Ni-Co-Cr alloys containing Cr less than 1/3 (e.g., Scientific Reports 6, 26179, 2016), the arguments against magnetism do not seem to be valid.

The authors carried out additional calculations simulating the paramagnetic state "We have conducted additional DFT calculations to evaluate the local magnetic moments of Cr, Co and Ni in the NiCoCr alloys in the paramagnetic state, based on the disordered local moment (DLM) model implemented in the EMTO-CPA package. Our results indicate that all the local moments of the atomic species are essentially zero.". This may very well be correct. However, it is not explicitly mentioned whether these additional calculations used static DLM or took into account the thermal spin fluctuations. At temperatures of interest, local magnetic moments are very unlikely to vanish like in a static DLM study.

2) SFE and plasticity.

They write "For example, the experimentally measured SFE of NiCoCr is usually a positive number around 20 mJ/m² [Acta Materialia 128, 292 (2017)], while most computational work assuming random solid solutions (even at room temperature) give a near-zero or negative SFE [Acta Materialia 134, 334 (2017), Nat. Commun., 9, 2381, (2018) and so on], which is problematic when adopted to interpret experimental results.". This statement suggests that the authors do not make difference between experimental and theoretical SFE. The experimental SFE is never measured directly and thus it reflects a different physical parameter than the one computed using infinitely large planar fault. This is especially the case in alloys with very small or slightly negative SFE.

3) Kinetics.

The question of reaching certain LCOs is very strongly dependent on kinetics. The authors admit that "the partially ordered systems should be accessible, provided the experiments are done with adequate ageing at appropriate temperatures.", "given enough kinetics the microstructure evolves towards the ground state" Without experimental verification, this remains a question that weakens the impact of the present work.

4) Connection to observed data.

The negative local SFE is expected to lead to phase transformation which was not observed by Gludovatz et al. Nat. Commun. 7, 10602 (2016) but reported in the work by Niu et al. Nat.

Commun. 9, 1363 (2018). The authors explicitly refer to the above work. However, Niu et al. discuss the magnetic effects and magnetic frustration in the present alloy (and CrMnFeCoNi). Magnetism is not considered here at all, meaning that those effects reported by Niu et al. cannot be captured by the present modeling.

In summary, the criticisms by both reviewers point towards a horizontal rather than a vertical development. Because of that and because of the above specific issues I cannot support the present work in journals like NM.

Reviewer #1:

This paper presents an atomistic study of solid-solution strengthening due to local chemical ordering (LCO) in a medium-entropy alloy of NiCoCr. The authors developed an EAM potential for the alloy and used this potential to investigate the impact of annealing temperature on LCO. They further studied effects of the LCO on planar fault energies as well as activation energies for dislocation detrapping.

Overall, the paper is topical and important. The modeling work has been carefully performed and described. The results are very rich, and they provide new insights into the relationship between chemical structures and dislocation activities in medium/high-entropy alloys. The paper can be recommended for publication in Nature Communications, provided the following comments are addressed adequately.

Reply:

We appreciate the reviewer's positive comments on our work.

Fig. 4b-e shows the reaction pathway calculation for a leading partial dislocation in a cylindrical cell with the axial direction along $\langle 112 \rangle$. This setup is also used to calculate the energy barriers shown in Fig. 5. However, the orientation of this simulation cell is different from that in Fig. 3 and Fig. 4a. To avoid possible confusion, the crystallographic orientations (e.g., along the axial and dislocation glide directions in Fig. 4c) should be clearly labelled.

Reply:

We thank the reviewer for this suggestion. We have now clearly labeled the crystallographic orientations in the revised Fig. 4.

It is not clear why the authors employed the string method, as opposed to the commonly-used nudged elastic band (NEB) method, to calculate energy barriers for dislocation detrapping. For the minimum energy path shown in Fig. 4b-e, what is the corresponding NEB result? It would be helpful to use the NEB method for verifying this result from the string method. In particular, the convergence criteria (i.e., image displacement and maximum number of iterations) used by the authors may not give a truly converged result, when the transition pathway is relaxed on a slowly changing landscape. In addition, given the rugged energy landscape governing the dislocation detrapping response in complex alloys, there are possibly multiple transition pathways between the same set of initial and final states. How did the authors deal with this kind of situation?

Reply:

For energy barrier calculations, we followed the suggestions made by Nöhning and Curtin [*Acta Materialia*, 128, 135-148, 2017; *Acta Materialia*, 158, 95-117, 2018] that the string method could be more robust for complex concentrated alloys. Nevertheless, we did verify by comparing the string method results with NEB results. Fig. R1 is an example directly comparing the

minimum energy paths (MEPs) calculated using the string method and the NEB method. As can be seen, the overall profiles of both MEPs are very similar to each other; the major saddle points coincide with each other very well, suggesting very good consistency.

Fig. R1 | Comparison between the minimum energy paths calculated by the NEB method and string method.

We also extensively tested the convergence criteria used in the string method. Overall, results obtained using the current criteria are comparable to well converged NEB calculations. For example, in the above figure, the NEB calculation was considered as converged only when the forces on each replica are less than 0.001 eV/\AA .

We agree with the reviewer that for complex rugged energy landscape, there may be multiple transition pathways between the same set of initial/final states. The transition rate of a *specific event* may require advanced methods such as finite-temperature string method, forward flux sampling etc. However, in this work, we are not focusing on a single event but rather on sampling the whole set of possible nanoscale segment detrapping events in the entire sample. For this purpose, string method calculations/NEB calculations over many randomly chosen events would draw a sample set that reflects the characteristics of the entire distribution. The calculated barriers can be further verified using Arrhenius plot. For example, Fig. R2 plots the natural logarithm of partial dislocation velocities ($\ln(v)$) against $1/(k_B T)$ in random solution samples subjected to a shear stress of 200 MPa. As can be seen, the fitted average activation energy in this case is 0.052 eV, which is close to the string-method calculated activation energy 0.085 eV (with a standard deviation of 0.0625 eV). On the other hand, the fitted intercept $\ln(A)$, where $A = v_0 \bar{d} \exp(\Delta S)$, is 7.8. As we already know \bar{d} and $\exp(\Delta S)$ (from the thermodynamic compensation rule in Section S9), the attempt frequency is estimated to be $2.1 \times 10^{12} \text{ s}^{-1}$, which is in the expected range from 10^{11} s^{-1} to 10^{13} s^{-1} . Thus, we believe our current method should be a reasonable approximation to the true activation energy distribution. We have added these verification and discussion in the revised SI.

Fig. R2 | Arrhenius plot of partial dislocation velocities ($\ln(v)$) vs. $1/(k_B T)$ in RSS. The applied shear stress is 200 MPa. The dislocation line length is ~ 30 nm and the glide distance is ~ 64 nm. Each data point was averaged over 5 different RSS samples. The Arrhenius-plot-informed average activation energy is 0.052 eV (the negative slope), which is very close to the string-method calculated activation energy 0.085 eV with a standard deviation of 0.0625 eV.

The MD results of edge dislocations in the Supporting Information are very interesting. However, the simulation setup is not clearly described. What is the simulation geometry? Did the authors prescribe velocity at the boundary layer of the simulation cell? A schematic illustration should be helpful. These details should be provided, as they may help understand why the applied load for detrapping a trailing partial nearly doubles that for a leading partial for T_a of 950K and 1350K. These results are likely caused by the displacement-controlled mode of loading and possibly affected by small in-plane sizes – in either x or y direction or both.

Reply:

We thank the reviewer for the positive comments on our simulations. We have added a schematic illustration on the simulation setup in the revised Supporting Information. Specifically, the simulation box has a geometry of $X[11\bar{2}] 52 \text{ nm} \times Y[111] 12.5 \text{ nm} \times Z[\bar{1}\bar{1}0] 111 \text{ nm}$. For shear deformation under constant strain rate, we assigned constant velocities to atoms in surface layers, which is a widely used procedure in MD simulations. The current simulation box has a large in-plane size of 52 nm (dislocation line direction) \times 111 nm (dislocation motion direction), so the box size effects should have been minimal.

The higher stresses needed to drive the trailing partial dislocations are caused by the relatively larger energy cost to eliminate the complex stacking fault (CSF) and create local antiphase boundaries (APB). This can be seen from differences between APBE and CSFE, i.e., APBE – CSFE is a larger value when compared to CSFE. For example, for $T_a = 1350$ K and $T_a = 950$ K, the differences between APBE and CSFE are 52.24 mJ/m^2 and 58.37 mJ/m^2 , respectively, while their CSFEs are only 9.11 mJ/m^2 and 22.58 mJ/m^2 , respectively. Thus, the average energy penalty to eliminate CSF is considerably higher than that to create CSF, resulting in higher stresses to drive trailing partial dislocations. In contrast, for $T_a = 650$ K, APBE – CSFE is 60.60 mJ/m^2 while the CSFE is 60.69 mJ/m^2 , thus the stresses needed to drive leading partial dislocation and trailing partial dislocation should be comparable to each other. However, for such lower T_a , the stresses to drive leading and trailing partial dislocation would also depend on the local Co-Cr domain orientations (see Fig. S12). We have added such discussion in the revised SI.

The Labusch theory has been widely used to understand the solid-solution strengthening effects in medium/high-entropy alloys. The trough model developed by Kocks and others has been often used to explain the temperature and strain-rate effects on solid-solution strengthening. What are the implications of the present work on the application of these classical theories to medium/high-entropy alloys?

Reply:

In the Labusch theory, strengthening is attributed to the collective interactions between many solute atoms and a dislocation, i.e., it is the favored statistical fluctuations in solute configuration that pin dislocation, rather than individual solute atoms as considered in Fleischer's model. Labusch's assumption is consistent with the atomistic mechanism of dislocation motion we uncover and demonstrate: we see that dislocations are pinned by favored local chemical environments (see Fig. 3 and Fig. 6 in main text for the wavy dislocation configurations) and the motion mechanism is nanoscale segment detrapping from these local favored chemical environments. However, Labusch's model also predicts a concentration dependent strengthening effect; this is out the scope of the current work and needs to be considered in future work.

In the trough model of Kocks and others, it was assumed that a dislocation can collect solute atoms from its surroundings and the solute atoms smeared out along the dislocation, creating a lower free energy state or bound state. Unlocking from such bound state was modeled in terms of bulge nucleation. This trough model has been shown successful in modeling the temperature and strain rate dependence of flow stress in solution hardened single crystals. However, the assumption in the trough model may not be consistent with our current findings. For example, the energy landscape of dislocation in our model system is more complicated than the simplified states (bound state and high free line energy state) assumed in the trough model. Also, the trough model can imply a large activation volume close to $1,000b^3$, while in our work the LCO is more localized and the activation volume is found to be on the order of $10^0 b^3 - 10^2 b^3$. However, we do not rule out the possibility that a modified version of the trough model could model the intermediate to high temperature dislocation behaviors in a complex concentrated system like our ternary model system.

Line 105, the authors state “This tendency is consistent with the equilibrium phase diagram to form Co-Cr intermetallic phase(s)”. Please provide a reference for this statement.

Reply:

We have cited the phase diagrams (also displayed here) in the revised text.

© ASM International 2006. Diagram No. 1600355

© ASM International 2006. Diagram No. 101102

© ASM International 2006. Diagram No. 900728

Reviewer #2:

Li et al systematically studied the local-chemical-order effect on the deformation behavior in NiCoCr medium-entropy alloys by largescale atomistic simulation. They developed a realistic interatomic potential for the NiCoCr system and demonstrated that the local chemical order can produce a wide range of planar fault energy, heighten the ruggedness of the energy landscape and raise activation barrier governing dislocation activities.

This manuscript is very nicely written and the reading is delightful. The atomistic simulation is pretty solid and extensive data have been provided (especially in SI). The following are my comments:

1) The authors insisted they developed the realistic interatomic potential for the NiCoCr systems (with extensive validation in SI). But as mentioned in the Methods, the spin-polarization effect was not considered in the potential fitting. That adds concerns to such empirical potential. I think the effect of spin-polarization is nontrivial, considering the large difference of lattice size between the non-spin-polarized NiCoCr (see Ref. 36) and spin-polarized NiCoCr (see Ref. 35,39,40). Also, the calculated intrinsic stacking fault energy are different with or without considering magnetism (see above reference). More importantly, I think the spin-polarization effect becomes more pronounced for their hybrid MC/MD simulation. That's because the local magnetic moment can be varied after each MC swaps and the accrual effect would be nontrivial after millions of swaps. I understand it's not easy to develop a highly optimized potential for multicomponent alloys, especially for those with magnetism. I would suggest looking further into those concerns.

Reply:

This reply addresses the concerns of both Reviewer #2 and #3 in regards to magnetic effects.

We stated that the Ni-Co-Cr potential developed was a “realistic potential” in the context that no realistic potential exists so far for multi-component HEAs (the published MD simulation papers had to use some “average atom” potentials), seriously hampering atomistic modeling of these new concentrated alloys. As detailed in the validation section in SI, our potential was developed based on non-magnetic DFT calculations and experimental inputs. Spin polarization was not explicitly considered. So in the revised ms we have decided to change the wording of “realistic potential” to “**empirical** potential”. Ours is a highly optimized atomistic model for equi-atomic multicomponent system, the best there is for HEAs, but the reviewers are right that it is not “realistic” to the point that it can describe magnetic transition and spin polarization effects.

1) In fact, atomistic models employing classic potential formalisms (e.g., the embedded-atom-method adopted here) never fully account for the electron spins. This is because of i) the limitation of the potential formalism: the goal is the construction of a potential that allows handling of large-scale simulations; but this is achieved at the expense of the accuracy of first-principles calculations. ii) The level of difficulty in accurately describing the magnetic states of the system. In practice, the magnetic effect is usually not explicit in such classic MD potentials and simulations. Even for pure Ni, with a Curie temperature of ~627K, MD simulations do not have the predictive power for its magnetic transition.

In general, atomistic models developed using the force-matching method will not be as accurate as DFT calculations. In our development of the Ni-Co-Cr potential, despite of our best effort, the average energy difference between the DFT data and the EAM potential is ~20 meV/atom (see SI). In the field of potential development, this margin of deviation is normally already considered a highly optimized interatomic potential for atomistic MD modeling. But the empirical potential could fail to capture the magnetic effects, if the resultant energy difference is only several

meV/atom. This is essentially inherent to the empirical potential development, rather than our potential fitting procedure itself.

2) Then the next question is “how large are the magnetic effects in NiCoCr”. Here we start our discussion from the random solution at this composition. Our spin-polarized DFT calculation shows a very small energy difference between the magnetic phase and the non-magnetic fcc phase, 2 meV/atom, and that between magnetic and non-magnetic hcp states is around 4 meV/atom (see Figure R3 and its caption for methods). The SFE of the magnetic state is slightly more negative than the non-magnetic state by a difference of $\sim 5\text{-}10\text{ mJ/m}^2$. Experimentally, according to an Oak Ridge group in *Scientific Reports* 6, 26179, 2016, the antiferromagnetism of Cr “frustrates” the ferromagnetism of NiCo. The magnetic ordering does not show up at this composition all the way down to 2 K. The system behaves like a paramagnetic material with a susceptibility like Pd metal.

Fig. R3 | Results of *ab initio* calculations of the energy differences between fcc and hcp NiCoCr at 0K. Equation of states of fcc and hcp CrCoNi phases in the non-magnetic (left panel) and the ferromagnetic (right panel) states. The energies are plotted with reference to the non-magnetic fcc NiCoCr structure at the ground state. In both cases, the hcp structure is found to have a smaller formation energy. The magnetic contribution is found to be small in the NiCoCr alloy.

But this 2016 experimental measurement also indicates that when the Cr content is lowered, magnetic ordering does become more obvious. When the alloy composition shifts to NiCoCr0.5 (Curie temperature rises to 250 K), our spin-polarized DFT calculations show that the energy difference between the magnetic and non-magnetic states becomes 15 meV/atom (and the volume difference is less than 2%). Therefore, even in this “obviously magnetic” case, the energy difference between the spin-polarized and non-polarized states is still relatively small, not up to the level that an EAM potential can resolve.

As pointed out by the referees, magnetic effects may increase in our modeling when our solution alloy develops increasing local (partial) chemical order at low ageing temperatures. To evaluate the magnetic contribution in these configurations (each 360 atoms) with various LCOs (Figure S7), we performed spin-polarized DFT calculations on them. The configuration with the largest compositional variations shows an energy difference that is similar to that of the NiCoCr0.5 case above.

We mention here that the mechanical behavior of interest is mostly in the temperature range from room-temperature down to liquid nitrogen temperature. The zero-K potential energy landscape we describe is actually the finite-temperature free energy landscape with the thermal contribution subtracted from it. This energy landscape is not really the one at 0 K in the presence of ferromagnetic states, as finite temperature would randomize most of the spins.

3) Having said all the above, we still have to admit that fundamentally, the EAM formalism is incapable of capturing complex magnetic effects at finite temperatures in a self-consistent manner. Some type of spin-dependent potential needs to be developed, but is beyond reach at present. As mentioned earlier, thus far EAM potentials have never been meant to monitor various degrees of magnetic ordering.

Therefore, in the revision, we follow the referees' suggestion to soften the claim. We have removed the wording "realistic" and now introduce the EAM potential as an empirical NiCoCr-like atomistic model that enables a parametric study of the trend of dislocation behavior. This atomistic model is designed to capture the typical features of HEAs and MEAs: multi-principal (equiatomic) constituents, moderate chemical interactions, similar atomic sizes, single phase but with varying degrees of local chemical order, etc. The model is meant to analyze the trend due to LCO and consequences on dislocation responses, rather than pinning down energy numbers or nailing down the various contributions from chemical, elastic or magnetic energy terms. The model still overcomes a major hurdle in atomistic studies of high-entropy alloys: it allows large-scale (e.g., millions of atoms MD simulations) modeling of dislocation activities, outside the realm of the more accurate first-principles DFT calculations. We have added these discussions in the revised SI.

2) As shown in Fig. 1, they indicated the aggregated Ni atoms in the nanoscale domains. This looks very interesting. But I also find other previous literature (Ref. 23 and 40) using DFT calculation revealed the different short-range order after MC simulation. I know the sample in DFT is indeed much smaller than that in present work and the MC steps is also very limited. Could the authors have a test to use their potential and MC methods in a small configuration with limited MC steps? This can help the validation of present work. Also, Ref. 34 has revealed the nanoscale clustering of oxygen in a BCC HEA by atom probe tomography. So if there is indeed the domains of aggregated Ni atoms as predicted in present work, the direct experimental characterization would provide very strong evidence.

Reply:

We tried lattice Monte Carlo simulations on small supercells. Specifically, 1,000 special quasi-random structures (SQS) with 108 atoms were first obtained. These SQSs all have zero Warren-

Cowley short-range-order parameters, suggesting random solid solutions. To satisfy the minimum image convention, all SQSs are then replicated in three directions, generating 1,000 bigger supercells each with 864 atoms. For limited MC steps (as used in the DFT calculations in Ref. 23 and Ref. 40), the average swaps per atom was limited to ~ 25 . The sampling temperature in MC simulations is 500 K and 1,000 independent MC simulations were carried out for statistical purposes. The results are shown in Fig. R4, in terms of scatter plot and marginal (kernel density) distribution for each type of atom-pair. As can be seen, small supercell and limited MC steps indeed result in differences from the well-converged MC simulations using our large samples. First, the calculated chemical short-range order parameters are highly scattered; very often one can see opposite (sign) chemical ordering in two different MC simulations. This points to the need for a large number of independent MC simulations to obtain a correct trend. Second, in terms of the *average* Warren-Cowley short-range-order parameters, we see negative values (favored pairing) for the Ni-Ni, Ni-Cr and Co-Cr pairs, while the Ni-Co, Co-Co and Cr-Cr pairs all show positive values: so these limited MC simulations are consistent with previous DFT+MC calculations on similarly small samples. In particular, all predict preferred Ni-Cr and Co-Cr local order, except that the slightly negative value was not reported for Ni-Ni.

As for experimental characterization, so far no one has aged NiCoCr for a long time at rather low temperatures to obtain nanoscale domains, which would then become detectable using atom probe tomography or other tools.

Fig. R4 | Chemical short-rang-orders (the Warren-Cowley parameter) obtained from lattice Monte Carlo simulations with limited swaps per atom (25 swaps/atom). Positive values correspond to a tendency of decreasing the number of i,j pair while negative values show a tendency to increase the number of i,j pair. The initial states of all samples are special quasi-random structures (SQS) for which the Warren-Cowley parameter for the 1st shell is all zero. The

SQS structures have 108 atoms and then was replicated along three directions by a factor of two to meet the minimum image convention. The sampling temperature was set as 500 K. 1,000 independent MC simulations were performed for statistical purpose.

In sum, I can't recommend this manuscript for the publication at Nature Communications. I would reconsider it if the authors can solve my questions and concerns.

Reviewer #3 (Remarks to the Author):

Report to NC

I have seen this work two times when it was submitted and resubmitted to Nature Materials. It is sad that the authors consider alternative journals/publicity instead of properly revising their work. I see very little changes here compared to the previous versions and therefore I am very sorry but I cannot be more positive about it for Nature Communications either.

Reply:

We have made an overhaul to the manuscript and provided much-needed insight into dislocation motion, its activated process (with activation parameters such as activation energy and activation volume), the nanosegment detrapping mechanism, and the strengthening derived from the mechanism (different from normal fcc) and due to LCO. This is a major expansion over the previous version.

We have also added a long discussion (as Section S12 in Supplementary Information) to discuss the magnetic effects. It also explains why the atomistic model we developed is an empirical potential, but critical for large-scale modeling of dislocation behavior in a concentrated solution.

With reference to my second report to NM:

1) The authors refer to Niu, C., et al. Nature Comm. 9, 1363 (2018) when presenting their additional spin fluctuation study. To the present referee this is very unclear and it would be more appropriate to give details about those additional tools, models and calculations and explain why no thermal spin fluctuations were accounted for here.

Reply:

Additional spin-polarized DFT calculations have been carried out to assess the effect of magnetism of NiCoCr alloys. We used the randomly populated super-cell approach to obtain the formation energies of fcc and hcp NiCoCr alloys, both in the magnetic state and in the non-magnetic state. The formation energies were averaged over 20 random configurations. For each calculation, the configuration (360 atoms) was geometrically optimized to reach the ground state

at 0K. Our results show that in either the magnetic or non-magnetic state, the hcp phase is energetically favored over the fcc phase. At the ground state, the magnetic contributions to the fcc and hcp phase are 2meV/atom and 4 meV/atom, respectively. Our results are consistent with Niu, C., et al, *Nature Comm.* 9, 1363 (2018), where the magnetic contributions to the hcp and fcc phases of equiatomic NiCoCr alloy were studied with the special quasirandom structure (SQS) method in the DFT treatment.

In both calculations, thermal spin fluctuations were not considered. In the ground state without the thermal effect, the contribution of spin ordering in the NiCoCr alloy is on the order of few meV/atom. If thermal spin fluctuations are explicitly considered, for instance, following Dong's approach (see Dong et al., *Scientific Reports* 12211, 2018), we estimate that the contributions owing to spin fluctuations are within a few meV/atom. This is inferred from the fact that the magnetic contribution due to thermal spin fluctuations to the intrinsic stacking fault energy of the Cantor alloy (FeMnCoCrNi) is ~ 5 mJ/m² at 900K (based on longitudinal spin-fluctuation calculations).

Because of the complicated spin states in the alloy as well as the relatively small magnetic contribution of electron spins, our atomistic model cannot explicitly capture the magnetic effect in the NiCoCr alloy. To fully account for the magnetic properties of the NiCoCr alloys, a more sophisticated atomistic model needs to be developed, which will be sought in our future research. Again, as stated in the reply to Referee #2 above, the goal of our EAM-based modeling work is a parametric and systematic study of dislocation behavior through large simulations, rather than an advance of first-principles treatment of small samples to rigorously resolve the relatively small magnetic contributions that are difficult to pin down.

2) The experimental SFE is quoted in the present work as well. No comments on its meaning and errors are given. Instead they associate the large (about 20 mJ/m²) experimental value to LCO. Might be very misleading. In addition, see my previous reports to the NM version of the present manuscript.

Reply:

We now only mention in passing that, in experimental studies of real-world NiCoCr, an SFE (20 mJ/m²) was inferred. In the revised version we no longer associate it specifically with the LCO in our model.

In short, the present work is a valuable attempt to revise the common picture behind high entropy in high entropy alloys. It is demonstrated on a model system without aiming for completeness and without direct verification.

Reply:

As discussed in the long reply to Referee 2, EAM model in classical MD simulations has its utility and advantages, but completeness or verified accuracy is unfortunately not one of them. Our revised paper emphasizes the former, and softens the conclusion associated with the latter.

In particular, we have added a long discussion (as Section S12 in Supplementary Information) to discuss the magnetic effects. It also explains why the atomistic model we developed is an empirical potential, but critical for large-scale modeling of dislocation behavior in a concentrated solution.

It is not considered for truly high-entropy cases where entropy could easily overcome the predicted LCO versus annealing temperature behavior. Although I would like to see this work published in some way but a substantial restructuring, reformulation and softening of the conclusions are needed to avoid confusion and misinterpretations.

Reply:

At temperatures $< 1,000$ K or so, so far there have been no HEAs that have high enough entropy to eliminate the possibility of LCO. Even the pioneering Cantor alloy with 5 elements/species decomposes with precipitates if annealed long enough. In other words, the single-phase solution is not the ground state but metastable. One can add more elements into the mix to hope for “high configurational entropy close to ideal solution”. But the more species, the more likely that there would be pairs not having zero enthalpy of mixing. In other words, it is not easy to find several elements that have similar sizes and all nearly zero hat of mixing. Mixing more elements also increase the chemical interactions along with increased mixing entropy, and truly high entropy cases exist only at high temperatures. Of course, one can rapidly quench a high-T configuration to room temperature to retain a random solution, but that is only one extreme case. During the normal lab practice of homogenization annealing (in our model, < 1650 K) or when the alloy is in service, the temperature is no longer all that high, so the local ordering tendency becomes difficult to overcome (which starts from short-to-medium range).

For the referee’s first report to NM last summer, we have made relevant changes in the manuscript and the referee has seen our point-by-point reply.

Reply to the second report to NM is given in the following.

Second report to NM

The authors have carefully considered the points raised by the two referees. They amended the manuscript at several places reaching a more readable and perhaps also more complete work. Nevertheless, the revision does not answer several key points. Here I list some:

1) Magnetism in Ni-Co-Cr system.

They claim “atomistic simulations using realistic potentials”, “we have spent considerable time and efforts developing a realistic EAM potential for the Ni-Co-Cr system”, and at the same time admit “We agree that our simulated NiCoCr is only a model system and have added a note to that effect following the reviewer’s comment. However, we also note that our specific case is representative”, “it is not imperative for the NiCoCr model to be a 100% accurate match for real-world NiCoCr in the lab”. These statements are contradictory. Furthermore, considering

that magnetism has been detected in Ni-Co-Cr alloys containing Cr less than 1/3 (e.g., Scientific Reports 6, 26179, 2016), the arguments against magnetism do not seem to be valid.

Reply:

We have changed the wording “realistic potential” into “empirical potential”. These questions are discussed in detail in the reply to reviewer #2 above. We have also added a long discussion (as Section S12 in Supplementary Information) to discuss the magnetic effects. It also explains why the atomistic model we developed, while an empirical potential, is critical for large-scale MD modeling of dislocation behavior in a concentrated solution.

The authors carried out additional calculations simulating the paramagnetic state “We have conducted additional DFT calculations to evaluate the local magnetic moments of Cr, Co and Ni in the NiCoCr alloys in the paramagnetic state, based on the disordered local moment (DLM) model implemented in the EMTO-CPA package. Our results indicate that all the local moments of the atomic species are essentially zero.”. This may very well be correct. However, it is not explicitly mentioned whether these additional calculations used static DLM or took into account the thermal spin fluctuations. At temperatures of interest, local magnetic moments are very unlikely to vanish like in a static DLM study.

Reply:

The paramagnetic state was studied with the static disordered local moment (DLM) method. The referee is correct that at elevated temperatures, due to magneto-volume effect and spin fluctuations, the local magnetic moments will not vanish and contribute to the free energy of the alloys. The spin fluctuations in the NiCoCr alloys are much more involved, which will be studied in our future work, for instance, by following Dong’s approach (Dong et al., Scientific Reports 12211, 2018). In the current work, however, we estimate that the contribution due to thermal spin fluctuations would be less than a few meV/atom (see our reply to question 1 above). Due to the limitations of our atomistic model, the spin effect was not explicitly taken into account.

2) SFE and plasticity.

They write “For example, the experimentally measured SFE of NiCoCr is usually a positive number around 20 mJ/m² [Acta Materialia 128, 292 (2017)], while most computational work assuming random solid solutions (even at room temperature) give a near-zero or negative SFE [Acta Materialia 134, 334 (2017), Nat. Commn., 9, 2381, (2018) and so on], which is problematic when adopted to interpret experimental results.”. This statement suggests that the authors do not make difference between experimental and theoretical SFE. The experimental SFE is never measured directly and thus it reflects a different physical parameter than the one computed using infinitely large planar fault. This is especially the case in alloys with very small or slightly negative SFE.

Reply:

We agree with the referee. In the revision we have removed this discussion, to avoid the confusion above. It is not advisable to expect a direct and perfect match between the calculated SFE from our empirical potential model and the experimentally measured SFE for NiCoCr.

3) Kinetics.

The question of reaching certain LCOs is very strongly dependent on kinetics. The authors admit that “the partially ordered systems should be accessible, provided the experiments are done with adequate ageing at appropriate temperatures.”, “given enough kinetics the microstructure evolves towards the ground state” Without experimental verification, this remains a question that weakens the impact of the present work.

Reply:

Chemical ordering has been reported/indicated in experiments on several HEAs. Examples include the Cantor alloy [Otto et al. *Acta Mater.* 112, 40–52 (2016)], the BCC TaNbHfZr HEA [Maiti & Steurer, *Acta Mater.* 106, 87–97 (2016)], the BCC TiZrHfNb HEA [Lei et al, *Nature*, 563, pages546–550 (2018)], the FCC CoCrFeNi HEA [He et al, *Scripta Mater.*, 126, 15-19,(2017)], the FCC NiCoCr MEA [Zhang et al, *PRL*, 18, 205501, (2017)], and the work using neutron diffractions by Ma et al. [Ma et al, *Scripta Materialia*, 144, 64-68, (2018)] etc. Also see a recent review on the metastability of HEAs [Wei et al, *J. Mater. Res.*, 33, 2924, (2018)]. In these works, either chemical short-range-order or composition decomposition has already been experimentally demonstrated. We emphasized this point and cited these references in the revised ms.

4) Connection to observed data.

*The negative local SFE is expected to lead to phase transformation which was not observed by Gludovatz et al. *Nat. Commun.* 7, 10602 (2016) but reported in the work by Niu et al. *Nat. Commun.* 9, 1363 (2018). The authors explicitly refer to the above work. However, Niu et al. discuss the magnetic effects and magnetic frustration in the present alloy (and CrMnFeCoNi). Magnetism is not considered here at all, meaning that those effects reported by Niu et al. cannot be captured by the present modeling.*

Reply:

The experimentally observed FCC to HCP phase transformation in NiCoCr is due to the lower (free) energy of the HCP phase. This energy difference between the FCC and HCP phases can be eliminated, as shown by Niu et al, in the CrMnFeCoNi HEA and other derivative ternary alloys, when magnetically frustrated Mn is present. This magnetic effect will not be captured in our model. However, Mn (hence this "magnetic effect") is absent in NiCoCr. For this alloy, Niu's results can be captured in our model. Specifically, they state that "*in CrCoNi, the hcp phase is consistently favored over fcc, even when the magnetic effects are turned off. The average energy of either phase is reduced by considering magnetism, but the difference between the average energies of hcp and fcc phases does not change significantly.*" This is also seen in our current DFT calculations, shown in Figure R3. Our present EAM model is consistent with the DFT results. In fact, this is discussed in the context of intrinsic stacking fault energy in the main text:

our optimized potential was able to reproduce the negative stacking fault energy of NiCoCr random solid solution even without considering the magnetic effect.

In summary, the criticisms by both reviewers point towards a horizontal rather than a vertical development. Because of that and because of the above specific issues I cannot support the present work in journals like NM.

Reply:

Regarding the vertical development, we stress again that our atomistic model is the first that enables a parametric study of the trend of dislocation behavior. This atomistic model is designed to capture the typical features of HEAs and MEAs: multi-principal (equiatomic) constituents, chemical interactions consistent with NiCoCr-type solutions, similar atomic sizes, single phase but with varying degrees of local chemical order, etc. Previous models fail to catch the chemical interactions, let alone dislocation behavior. The model shows a clear trend due to LCO and consequences on dislocation responses, providing much-needed insight into dislocation motion, its activated process (with activation parameters such as activation energy and activation volume), the nanosegment detrapping mechanism, and the strengthening derived from the mechanism (different from normal fcc) and due to LCO. This series of development has, in our opinion, reached the NC level, if not NM. The new potential/model also overcomes a major hurdle in atomistic studies of high-entropy alloys: it allows large-scale (e.g., millions of atoms simulation) MD modeling of dislocation activities, outside the realm of the more accurate first-principles DFT calculations. A different line of vertical development indicated by the referee, in terms of pinning down energy numbers and various contributions from chemical, elastic or magnetic energy terms, is not the strength of EAM modeling and must await future theoretical work.

Reviewers' Comments:

Reviewer #1:

Remarks to the Author:

The authors have provided a comprehensive response to my review comments. They have well-addressed my major concerns from the previous round of review. In addition, they have changed the wording of “realistic potential” to “empirical potential” and thus given a more realistic assessment of contributions of their work. Indeed, their empirical NiCoCr-like atomistic model enables a parametric study of the trend of dislocation behavior. The results shed new lights onto the strengthening effects in concentrated multi-component alloys. In my opinion, this work has met the standard for publications in Nature Communications. Hence I am happy to recommend the publication of this work in Nature Communications.

Reviewer #2:

Remarks to the Author:

In the reply letter and revised manuscript, the authors have solved my first comment on the EAM potential. But my second concern remains, as below:

Fig. R4 shows the MC results of small samples using EAM potential, and interestingly the measured chemical short-range order are scattering. However, such CSRO values are not consistent with previous DFT results, especially for those around Cr atoms, regardless of the scattering nature. For example, the EAM potential simulation gives the value for Cr-Cr between 0 and 0.10 (with the average around 0.05). But Ref.23 presented the DFT simulation results of 0.42. Such difference is nontrivial.

Moreover, even at relatively high temperature (e.g. 950K), we can observe the clustering of Ni atoms in Fig. 1. If this can be validated by experimental techniques, it would significantly elevate the importance of this work.

Reviewer #3:

Remarks to the Author:

Second report to NC

Although I still have some concerns with the present manuscript, reading all reports and replies, I tend to change my verdict and support this work for publication in NC if the authors fix the following issues within the latest version.

- 1) The authors continue talking about HEAs in the abstract. This must be changed as the present work is not about HEAs. The concept could in principle be extended and that should be discussed in the conclusion section, but in the abstract they should talk about the ternary (medium entropy) alloy.

- 2) In the reply they say that the estimated magnetic effect is 15 meV/atom, which they consider to be small. That is not true. Such energy change can substantially modify the formation energy of a defect (stacking fault) especially if the magnetic effects are sensitive to the local structure around the defect. The authors should not deny and blur the impact of magnetism. They should simply admit that they have no way to account for it and leave the question open. In that respect this work remains valid for a hypothetical non-magnetic CrCoNi alloy. Nothing is wrong with that just admit it.

- 3) On page 7, they write “.. beyond 3b slip, the average CSFE and APBE decay to values similar to the RSS. This suggests that shear larger than 3b can destroy most of the LCO on the neighboring planes, leading to a slip plane softening mechanism⁴⁸ responsible for the experimentally observed planar slip. Plastic deformation is then a practical route to convert LCO to quasi-random state.” This is very important and to some extent it contradicts to what is claimed in the abstract “... All these open a vast playground not accessible to ground-state SSs ...”. To avoid misunderstanding, please state in the abstract that the disclosed LCO and its impact on plasticity is expected to be important for the initial deformations, and gradually vanishes with increasing dislocation activity.

- 4) Previously I criticized the lack of vertical developments within the present manuscript. Here I specifically ask the authors to comment the relation of their work to the recent publication by Ding et al. (Ref. 43 in the latest manuscript). In the main text they cite that work as “In other words, an FCC HEA/MEA can take a CSFE value out of a wide range⁴³ and does not necessarily have the low SFE anticipated...”. A more detailed comparison of the present findings and those reported by Ding et al. is needed especially that both works emphasize the critical role of local chemical fluctuations in multi-principal element alloys. To be clear, in my opinion the present work goes far beyond the

previous work by Ding et al. (and thus deserves publication in a good journal), but due to the overlap one cannot simply ignore discussing those results in more details.

Reviewers' comments:

Reviewer #1 (Remarks to the Author):

The authors have provided a comprehensive response to my review comments. They have well-addressed my major concerns from the previous round of review. In addition, they have changed the wording of “realistic potential” to “empirical potential” and thus given a more realistic assessment of contributions of their work. Indeed, their empirical NiCoCr-like atomistic model enables a parametric study of the trend of dislocation behavior. The results shed new lights onto the strengthening effects in concentrated multi-component alloys. In my opinion, this work has met the standard for publications in Nature Communications. Hence I am happy to recommend the publication of this work in Nature Communications.

Reply: We thank the reviewer for the positive comments on our revisions.

Reviewer #2 (Remarks to the Author):

In the reply letter and revised manuscript, the authors have solved my first comment on the EAM potential. But my second concern remains, as below:

Fig. R4 shows the MC results of small samples using EAM potential, and interestingly the measured chemical short-range order are scattering. However, such CSRO values are not consistent with previous DFT results, especially for those around Cr atoms, regardless of the scattering nature. For example, the EAM potential simulation gives the value for Cr-Cr between 0 and 0.10 (with the average around 0.05). But Ref.23 presented the DFT simulation results of 0.42. Such difference is nontrivial.

Reply: The different magnitude of CSRO parameter, as predicted by the previous DFT calculations and our EAM model, may result from different treatments of magnetism in the two models. In previous DFT calculations (both ref. 23 and 32), the magnetic states of NiCoCr with LCO developed at finite temperatures were treated with ground-state spin-polarized calculations as if they were at 0 K, without considering spin fluctuations (both thermal and static fluctuations). In our EAM model of the alloy, the spin polarization was not explicitly considered in a scenario that could be regarded as roughly non-magnetic. Both are not accurate descriptions of the true magnetic state of this alloy, as adamantly argued by Reviewer #3.

Such magnetic effects are more significant when the local configuration has a composition (LCO) considerably different from the global average NiCoCr. Specifically, the energy difference in DFT calculations with/without magnetism is only a few meV/atom for the equi-atomic composition, but increases to ~15 meV/atom for NiCoCr0.5. This energy difference from magnetic ordering need to be taken into account together with that due to local chemical ordering. As a result, it is expected that calculations with different assumptions of magnetic effects would result in differences in the magnitude of CSRO parameters.

From our MC simulations on small samples, we see that our EAM potential predicts a favored Ni-Ni pairing (did not show up in previous DFT calculations Ref. 23) and relatively weaker Ni-Cr and Co-Cr pairing (as compared to previous DFT results in Ref. 23). As a result of these, the Cr-Cr CSRO value is less positive when compared to the DFT results in Ref. 23. Nevertheless, both our EAM model and previous DFT results (Ref. 23) captured the tendency for Co-Cr pairing, which is consistent with the equilibrium phase diagrams as cited in our main text.

We have added more discussions on this issue in the revised main text, highlighted on p. 4.

Moreover, even at relatively high temperature (e.g. 950K), we can observe the clustering of Ni atoms in Fig. 1. If this can be validated by experimental techniques, it would significantly elevate the importance of this work.

Reply: We thank the reviewer for this suggestion. We will collaborate with experimental groups to look into this. Specifically, in the lab one can age the alloy at 950K for long times, and try to detect the Ni local clustering in future work.

Again, as we have pointed out above, empirical potentials are not meant to accurately capture energy or order parameter values. Both our EAM and previous DFT models used assumptions, and both need to be validated using experiments if one aims to be quantitatively accurate. In terms of calculations, a rigorous treatment of magnetic effects requires a vertical development in the future, as pointed out by Reviewer #3.

Reviewer #3 (Remarks to the Author):

Second report to NC

Although I still have some concerns with the present manuscript, reading all reports and replies, I tend to change my verdict and support this work for publication in NC if the authors fix the following issues within the latest version.

1) The authors continue talking about HEAs in the abstract. This must be changed as the present work is not about HEAs. The concept could in principle be extended and that should be discussed in the conclusion section, but in the abstract they should talk about the ternary (medium entropy) alloy.

Reply: We have revised the abstract and conclusion accordingly, highlighted in the revised main text.

2) In the reply they say that the estimated magnetic effect is 15 meV/atom, which they consider to be small. That is not true. Such energy change can substantially modify the formation energy of a defect (stacking fault) especially if the magnetic effects are sensitive to the local structure around the defect. The authors should not deny and blur the impact of magnetism. They should simply admit that they have no way to account for it and leave the question open. In that respect this

work remains valid for a hypothetical non-magnetic CrCoNi alloy. Nothing is wrong with that just admit it.

Reply: We have revised the abstract, the SI and the main text to further clarify that our EAM model is for non-magnetic CrCoNi alloy system. The influence of possible magnetic effects is now also explicitly discussed in the revised main text, as highlighted in the section of “Variable LCOs in samples processed at different temperatures”.

3) On page 7, they write “.. beyond 3b slip, the average CSFE and APBE decay to values similar to the RSS. This suggests that shear larger than 3b can destroy most of the LCO on the neighboring planes, leading to a slip plane softening mechanism⁴⁸ responsible for the experimentally observed planar slip. Plastic deformation is then a practical route to convert LCO to quasi-random state.” This is very important and to some extent it contradicts to what is claimed in the abstract “... All these open a vast playground not accessible to ground-state SSs ...”. To avoid misunderstanding, please state in the abstract that the disclosed LCO and its impact on plasticity is expected to be important for the initial deformations, and gradually vanishes with increasing dislocation activity.

Reply: We thank the reviewer for this comment. We have revised the abstract (see highlighted new wording) to reflect that the impact of LCO is gradually reduced upon large plastic deformation. Note, however, that 3b shear on each and every atomic plane, to make the alloy “random” w/o LCO, would correspond to a large strain (the shear strain is on the order of 300%). In other words, to make the LCO in an HEA/MEA vanish one would need severe plastic deformation.

4) Previously I criticized the lack of vertical developments within the present manuscript. Here I specifically ask the authors to comment the relation of their work to the recent publication by Ding et al. (Ref. 43 in the latest manuscript). In the main text they cite that work as “In other words, an FCC HEA/MEA can take a CSFE value out of a wide range⁴³ and does not necessarily have the low SFE anticipated...”. A more detailed comparison of the present findings and those reported by Ding et al. is needed especially that both works emphasize the critical role of local chemical fluctuations in multi-principal element alloys. To be clear, in my opinion the present work goes far beyond the previous work by Ding et al. (and thus deserves publication in a good journal), but due to the overlap one cannot simply ignore discussing those results in more details.

Reply: We have added more detailed description/discussion on the work of Ding et al. (now Ref. 32), highlighted on p.4 and p. 6 in the revised main text.

REVIEWERS' COMMENTS:

Reviewer #2 (Remarks to the Author):

The response letter and revised manuscript have addressed my previous comments satisfyingly. Thus I am happy to recommend it for publication.

Reviewer #3 (Remarks to the Author):

The revised manuscript comes close to a version that could be accepted for NC. However, my feeling is that the authors still try to over-emphasize the impact of their work.

In particular, although they do not work with high entropy alloys (in the present manuscript) they still start their abstract with "High-entropy alloys (HEAs) ...".

Second, in the abstract they argue that "LCO heightens the ruggedness of the energy landscape and raises activation barriers governing dislocation activities. This not only influences the selection of dislocation pathways in slip, faulting, twinning, and martensitic transformation, but also increases the lattice friction to dislocation motion via a new mechanism of nanoscale segment detrapping that elevates the mechanical strength. Severe plastic deformation gradually reduces the LCO towards random SS." Furthermore, in the reply they mention that 300% lattice strain could remove the LCO turning the system "random". There is no direct evidence in the manuscript that could support this point. As demonstrated here, LCO rises the activation barriers and increases the lattice friction to dislocation motion. Then as soon as LCO is removed along some dislocation pathways new low-barrier paths will open which will be determinative for the mechanical strength rather than the LCO part of the matrix. In short, I am not convinced that LCO indeed changes the mechanical strength upon regular strain-stress regime and its impact vanishes only upon severe plastic deformation.

Reviewer #2 (Remarks to the Author):

The response letter and revised manuscript have addressed my previous comments satisfyingly. Thus I am happy to recommend it for publication.

Reply: We thank the reviewer for the positive comments on our revisions.

Reviewer #3 (Remarks to the Author):

The revised manuscript comes close to a version that could be accepted for NC. However, my feeling is that the authors still try to over-emphasize the impact of their work.

In particular, although they do not work with high entropy alloys (in the present manuscript) they still start their abstract with “High-entropy alloys (HEAs) ...”.

Reply: Editor Zoppi has already fixed this, in the first sentence of the revised abstract. The key identifier for HEA is multi-principal-element, not "how many elements and how high the configurational entropy". People continue to use the jargon "high-entropy" even when they know the configuration entropy is not all that high, and even when the alloy contains fewer elements than 5, or when there are more phases than the base solid solution and when the composition is far away from equiatomic. An example is that of Li et al., *Nature*, 534, 227–230, 2016.

Second, in the abstract they argue that “LCO heightens the ruggedness of the energy landscape and raises activation barriers governing dislocation activities. This not only influences the selection of dislocation pathways in slip, faulting, twinning, and martensitic transformation, but also increases the lattice friction to dislocation motion via a new mechanism of nanoscale segment detrapping that elevates the mechanical strength. Severe plastic deformation gradually reduces the LCO towards random SS.” Furthermore, in the reply they mention that 300% lattice strain could remove the LCO turning the system “random”. There is no direct evidence in the manuscript that could support this point. As demonstrated here, LCO rises the activation barriers and increases the lattice friction to dislocation motion. Then as soon as LCO is removed along some dislocation pathways new low-barrier paths will open which will be determinative for the mechanical strength rather than the LCO part of the matrix. In short, I am not convinced that LCO indeed changes the mechanical strength upon regular strain-stress regime and its impact vanishes only upon severe plastic deformation.

Reply: In this work, we explicitly show that the LCO influences the initial material strength; the increase of strength due to LCO has also been experimentally observed, such as those in the cited references [34,56] in the main text. Those findings may help convince the referee.

Plastic deformation well beyond the yielding point is much more complicated than the referee's over-simplified picture. Very often, dislocations see obstacles such as grain boundaries and forest dislocations, so they do not always run on a fixed atomic plane. In fact, there is often work hardening (or dislocation interactions). Plastic flow spreads onto other planes where there are always LCO. In this case, LCO obviously influences the flow stress. Even if an atomic plane

becomes random with repeated dislocation slip, part of it can easily recover LCO by dislocations cutting through the atomic plane, as illustrated in the figure below. As can be seen, while dislocation slip on the atomic plane eliminates LCO (red), subsequent dislocation slip cutting through the atomic plane (orange) implants new LCO from neighboring parallel planes. Thus, the complete elimination of LCO on a specific atomic plane is not as easy as the referee thought, i.e., only when most of the atomic planes lose LCO, which can only be achieved by severe plastic deformation, can cutting dislocations stop recovering the LCO. Overall, plastic deformation reduces the LCO of the entire sample, but dislocation activities are also activated in a variety of places depending on the stress tensor, the loading conditions, the sample crystallography and orientations, cross-slip, etc., preventing premature strain localization and strain softening.

In fact, if an already-sheared plane always remains the low-barrier path for dislocations to concentrate on exclusively, that means severe strain softening and then the material would fail on that plane at very low macroscopic strain. Obviously this does not happen: any normal metal can sustain plastic strain on the order of 50%.